# Paleocene-Eocene age glendonites from the Mid-Norwegian Margin - Indicators of cold snaps in the hothouse?

Madeleine L. Vickers[1*], Morgan T. Jones[1], Jack Longman[2], David Evans[3,4,†], Clemens V. Ullmann[5], Ella Wulfsberg Stokke[1], Martin Vickers[6], Joost Frieling[7], Dustin T. Harper[8], Vincent J. Clementi[9] & IODP Expedition 396 Scientists[10]

[1] Centre for Planetary Habitability (PHAB), University of Oslo, P.O. Box 1028 Blindern, 0315 Oslo, Norway

[2] Department of Geography and Environmental Sciences, Northumbria University, Newcastle-upon-Tyne, UK

[3] Institute of Geosciences, Goethe University Frankfurt, Altenhöferallee 1, 60438 Frankfurt am Main, Germany

[4] FIERCE, Frankfurt Isotope & Element Research Center, Goethe University Frankfurt, Altenhöferallee 1, 60438 Frankfurt am Main, Germany

[5] Camborne School of Mines, University of Exeter, Penryn Campus, Penryn TR10 9FE, UK

[6] Department of Chemistry, UCL, 20 Gordon Street, London WC1H 0AJ, UK

[7] Department of Earth Sciences, University of Oxford, Oxford, United Kingdom

[8] University of Utah, Dept. of Geology & Geophysics, 115 S 1460 E, Salt Lake City, UT 84112, U.S.A.

[9] Department of Marine and Coastal Sciences, Rutgers University, New Brunswick, NJ 08901, U.S.A.

[10] International Ocean Discovery Program, Texas A&M University, College Station TX 77845 U.S.A. Email: Expedition_396_Participants@iodp.tamu.edu

*m.l.vickers@geo.uio.no

[†] Now at: School of Ocean and Earth Science, University of Southampton, Southampton, SO14 3ZH, UK

**Keywords:** PETM; ikaite; Paleogene; International Ocean Discovery Program (IODP); JOIDES Resolution; Expedition 396; Sites U1567, U1568, U1569, U1570.

**Abstract**

The International Ocean Discovery Program (IODP) Expedition 396 to the mid-Norwegian margin recovered >1300 m of pristinely preserved, volcanic ash-rich sediments deposited during the late Paleocene and early Eocene, from close to the centre of the North Atlantic Igneous Province (NAIP).

Remarkably, many of these cores contain glendonites, pseudomorphs after the purported cold-water
mineral ikaite, from sediments dated to the late Paleocene and early Eocene. These time intervals
span some of the hottest climates of the Cenozoic, including the Paleocene-Eocene Thermal Maximum
(PETM). Global deep ocean temperatures are not thought to have dropped below 10 °C at any point
during this time, making the occurrence of supposedly cold-water (near-freezing temperature)
glendonite pseudomorphs seemingly paradoxical. This study presents a detailed sedimentological,
geochemical, and microscopic study of the IODP Exp. 396 glendonites, and presents an updated model
for the ikaite-to-calcite transformation for these glendonites. Specifically, we show that early
diagenesis of basaltic ashes of the NAIP appear to have chemically promoted ikaite growth in the
sediments in this region. Together with existing knowledge of late Paleocene and early Eocene
glendonites from Svalbard to the north, and early Eocene glendonites from Denmark to the south,
these new glendonite finds possibly imply episodic, short-duration, and likely localised cooling in the
Nordic Seas region, which may have been directly or indirectly linked to the emplacement of the NAIP.
**1. Introduction**
Glendonites are pseudomorphs after the mineral ikaite, a metastable, hydrated form of calcium
carbonate ($CaCO_3 \cdot 6H_2O$), which is found today growing in a range of environments (alkaline lakes, sea
ice, concrete, in estuarine settings, deep marine sediments and more; Council and Bennet, 1993;
Buchardt et al., 2001; Dieckmann et al., 2008; Boch et al., 2015; Zhou et al., 2015; Schultz et al., 2022;
2023a). Certain physical and chemical conditions are necessary in order for ikaite to precipitate
preferentially over the more stable anhydrous $CaCO_3$ polymorphs (calcite, aragonite, vaterite). These
parameters may include low temperatures, high alkalinity, and a range of possible chemical inhibitors
of the anhydrous polymorphs (e.g. high aqueous Mg, phosphate, and/or sulphate concentrations;
certain dissolved organic compounds) (Council and Bennett, 1993; Buchardt et al., 2001; Zhou et al.,
2015; Purgstaller et al., 2017; Stockmann et al., 2018; Whiticar et al., 2022). Whilst laboratory studies
have successfully, if fleetingly, precipitated ikaite at warm temperatures (≤ 35 °C), cold water
temperatures (< 10 °C) are much more favourable for the precipitation, growth, and longevity of this
mineral (Purgstaller et al., 2017; Stockmann et al., 2018; Tollefsen et al., 2020; Vickers et al., 2022). In
nature, ikaite has not been discovered growing above 7 °C (Suess et al., 1982; Dieckmann et al., 2008;
Zhou et al., 2015; Boch et al., 2015; Stockmann et al., 2022).
Glendonites are found throughout the geological record in marine sedimentary settings, sometimes
associated with glacial deposits (e.g. Kemper, 1987; Alley et al., 2020), in sediments deposited during
periods of both icehouse and greenhouse climate (Rogov et al., 2021; 2023). The occurrence of
glendonites in Mesozoic marine sediments has fuelled the debate about whether ephemeral polar ice
sheets waxed and waned during this long-term global greenhouse climate (e.g. Kemper, 1987; de Lurio
and Frakes, 1999; Rogov et al., 2017; Vickers et al., 2019; Merkel and Munnecke, 2023). Across the
Cenozoic sedimentary record, glendonites are found at increasingly broader latitude ranges through
time, coincident with the growth of polar ice caps and global climate cooling (Rogov et al., 2021).
However, reports of glendonites – including the largest ever discovered on Earth – from successions
deposited in the Nordic Seas region during the early Paleogene greenhouse have been a topic of
controversy in the paleoclimate community due to the apparent mismatch with concurrent
paleotemperature proxies (Huggett et al., 2005; Spielhagen and Tripati, 2009). Deep ocean bottom
water temperatures were >10 °C across the globe, including the North Atlantic, during much of the
Paleocene and Early Eocene (Zachos et al., 2008; Dunkley-Jones et al., 2013; Westerhold et al., 2020;
Meckler et al. 2022), such that the presence of glendonites in marine sedimentary sequences during
this time appears paradoxical. Whilst the successful synthesis of ikaite at warm (≤ 35 °C) temperatures
in laboratory conditions raises the possibility of ikaite/glendonite formation at much warmer
temperatures than modern-day natural ikaites (Purgstaller et al., 2017; Tollefsen et al., 2020), the
conditions under which this was achieved in the laboratory are unlike marine natural settings (e.g.,
compared to modern marine ikaite-bearing sites those precipitation experiments were characterised
by DIC concentrations at least an order of magnitude higher than typical pore water profiles and $\Omega_{calcite}$
>100, far in excess of that typically found in natural settings; Zhou et al., 2015).
In addition, clumped isotope temperature reconstructions for ikaite precipitation/transformation
temperatures from a Danish early Eocene succession suggest cold formation (or perhaps
transformation) conditions (0 - 9 °C) for the parent ikaite  (Vickers et al., 2020). Given that the ikaite
minerals grew from pore waters in the shallow subsurface (e.g., Zhou et al., 2015; Schultz et al., 2022),
this suggests that local bottom water temperatures must have been at least this cold (Vickers et al.,
2020), since temperature increases with burial depth. Reconstructed sea surface temperatures from
biomarkers for the North Sea area (Denmark) also suggest that short-term cooling events of
magnitude c. 5 - 7 °C (down to SSTs of 15 °C or lower) may have punctuated the late Paleocene to
early Eocene (Stokke et al., 2020a; Vickers et al., 2020; summarised by Jones et al., 2023). While the
potential drivers of intermittent cooling episodes have been touched upon by previous studies
(Schoon et al., 2015; Stokke et al., 2020a; Vickers et al., 2020), the causes, spatial extent, and
recurrence times of such events remains unresolved.
The International Ocean Discovery Program (IODP) Expedition 396 to the Norwegian continental
margin (August – September 2021) recovered numerous glendonites in  ash-bearing  deposits  dated

as latest Paleocene and earliest Eocene, including within those sediments deposited during the Paleocene-Eocene Thermal Maximum (PETM) hyperthermal event (Berndt et al., 2023; Planke et al., 2023a). Deep ocean bottom water temperatures may have reached 15 °C or more during the 150 – 200 kyr-long PETM hyperthermal (Röhl et al., 2007; Zachos et al., 2008; Murphy et al., 2010; Dunkley Jones et al., 2013; Westerhold et al., 2020). This makes the discovery of pseudomorphs after a mineral favoured by near-freezing conditions in shallow epicontinental seas truly remarkable if the age of their formation and recrystallisation coincided with the depositional age of the host sediments. If the parent ikaite grew during the PETM or the hothouse earliest Eocene climate (as was the case for the well-studied Fur Formation glendonites; Vickers et al., 2020), this raises questions about regional seaway connectivity and stratification in the Nordic Seas, as an extreme thermocline within these shallow seas is one way to reconcile cold bottom water and warm surface water proxies. The presence of glendonites in Nordic Seas strata could imply that current global and regional temperature reconstructions based on biogenic carbonates and lipid biomarkers do not represent the full spectrum of climate variability in the early Paleogene Northern Hemisphere.

This study documents and describes the glendonites recovered from IODP Exp. 396 cores and their stratigraphic distribution, using microscopic and geochemical analyses along with sedimentological data to elucidate the timing, climate, and chemical regimes that facilitated their formation.

**2. Materials and Methods**

*2.1 Geological setting and sampling*

IODP Expedition 396 drilled 21 boreholes along the mid-Norwegian continental margin during August and September 2021, recovering igneous and sedimentary rocks ranging from lava flow fields to hydrothermal vent complexes, including thick successions of upper Paleocene and lower Eocene strata (Fig. 1A). Ship-board sedimentary logging was followed by a more detailed high-resolution study of recovered cores stored at MARUM, Bremen, under refrigerated conditions (c. 4 °C). Paleocene-Eocene sedimentary successions were cored along the Modgunn and Mimir transects (Fig. 1B and C).

Boreholes from the Modgunn transect (Sites U1567-U1568) span the crater of a Paleogene hydrothermal vent complex close to the Vøring transform margin (Fig. 1B; Planke et al., 2023a,b). These vents formed due to the violent expulsion of volatiles generated by contact metamorphism of organic-rich sediments around igneous intrusions, and from magmatic degassing (e.g. Svensen et al., 2004). The Modgunn crater is approximately 80 m deep compared to the paleo-seafloor, with a 200-240m-wide feeder system that is rooted in a series of nested sill intrusions (Berndt et al., 2023). The

vent infill consists predominantly of laminated diatom-rich mud, mudstones, and ash layers that
rapidly accumulated to fill the bathymetric depression. The strata exhibit a negative $\delta^{13}C$ excursion
and the first and last occurrence of the biostratigraphic marker taxa *Apectodinium augustum* and
*Hemiaulus proteus* (Fig. 2). Based on these biostratigraphic and chemostratigraphic constraints,
Berndt et al. (2023) showed the vent formed just before the PETM onset and was completely filled
within the duration of the PETM CIE.
The Mimir High (Sites U1569-U1570) is a borehole transect through a marginal high on the Vøring
Transform Margin (Fig. 1C; Planke et al., 2023a,c). Uplift and erosion of the marginal high has removed
any basalt cover (Berndt et al., 2001), providing a window to access stratigraphic successions below
the breakup volcanism. The strata dip to the north, so a transect of boreholes was designed to provide
a composite late Paleocene – early Eocene section with overlap between each hole. These strata
consist of a mix of mud, mudstones, carbonates, igneous units, and ash layers (Planke et al., 2023a,c).
For three Holes (U1569A, U1570A and U1570D), on board biostratigraphy showed the presence of the
PETM-marker taxon *Apectodinium augustum* (Planke et al. 2023c) (Fig. 2).
Core sampling, including the collection of glendonites, was undertaken in April 2022 and high-
resolution sediment logging of the glendonite-bearing core sections was undertaken in August 2022,
at MARUM, University of Bremen, Germany.
At each hole, interstitial water (IW) samples were taken at intervals of ~3 m of sediment in the upper
50 m, ~1 sample every 9.5 m for the lower parts of the cored sediment. Standard IODP methods for
IW extraction were used at all sites. Following whole-core recovery to the catwalk, full round samples
were collected, sealed and transferred to the shipboard chemistry lab, where sediment exteriors were
carefully removed to reduce potential contamination from drilling fluids. The samples were
individually 'squeezed' - placed into a Carver press and subjected to 35,000 lb (15,875 kg) force.
Squeezed fluid was then filtered through a Whatman No. 1 filter (11 µm) and 0.5 mL was discarded.
The remaining fluids were collected in acid-cleaned syringes after filtering through 0.45 µm
polyethersulfone membranes, and split into aliquots. All analyses of the collected IWs were completed
following the standard shipboard methods of the R/V JOIDES Resolution (Planke et al., 2023d), and
are published in Planke et al. (2023a) and (2023b).
*2.2 Carbon isotope analysis*
A selection of bulk rock sub-samples were selected for carbon isotope analysis, in order to provide
new chemostratigraphic data for certain of the Mimir holes, and to complement the existing $\delta^{13}C_{org}$
dataset for the Modgunn transect (Berndt et al., 2023). Dried bulk rock samples were powdered and
decarbonated using 1 M HCl for 72 h, including a brief (c. 1 h) heating step at 50 °C during the
acidification process to remove any siderite that has been identified as occasionally present in these
sediments. The samples were then cleaned using deionised water and were oven dried at 50 °C before
being re-homogenised. Analyses of squeeze cake samples from the IW extraction procedure were
performed using a Thermo Fisher Scientific Flash Elemental Analyzer, coupled to a Thermo Fisher
Scientific DeltaV Isotope Ratio Mass Spectrometer at the CLIPT Lab, University of Oslo. An expanded
dataset from the samples collected in Bremen were analysed using a Thermo Fisher DeltaV Advantage
Mass Spectrometer at the School of Ocean & Earth Science & Technology (SOEST) at the University of
Hawaii at Mānoa. Each sample was run in duplicate to test reproducibility, which was < 0.06‰ for
both sample sets.
*2.3 Powder X-Ray Diffraction (PXRD)*
Powder X-ray diffraction (PXRD) was carried out on dried, powdered bulk sub-samples of five
glendonites from the Exp. 396 cores, using a Stoe StadiP transmission (thin-foil) diffractometer with a
copper anode at 30 mA, 40 kV and a germanium 111 monochromator to produce Kα1 X-rays. The
diffracted beam was collected by an 18° 2θ Dectris Mythen1K silicon strip detector. Samples were
sandwiched between two thin cellulose acetate discs and mounted in a holder set to spin continuously
during data collection. Data sets were scanned from 5 to 65° 2θ stepping at 0.5° and 20 s/step. The
resultant raw data has a step of 0.015° 2θ. Machine alignment was monitored using an NBS silicon
standard. Phase analysis was undertaken using Bruker's 'Eva' program (Gates-Rector and Blanton,
2019) interfaced with the Powder Diffraction File provided by the International Centre for Diffraction
Data.
*2.4 Microscopy*
Four polished thin sections were made from three glendonites (those in Cores U1569A-19R-2,
U1570A-15R-1, and U1567C-11X-1), and examined using a light microscope to compare to published
data on glendonite microfabric and look for different carbonate growth phases. A Hitachi SU5000 FE-
SEM, equipped with a Bruker EDS was used to examine porosity and spatial chemical variation across
the sub-samples.
*2.5 Laser Ablation Inductively Coupled Plasma Mass Spectrometry (LA ICP-MS)*
Laser ablation trace element analyses were performed on polished thin sections of glendonites from
Cores U1569A-19R-2, U1570A-15R-1, and U1567C-11X-1 at the Frankfurt Isotope and Element

184 Research Center (FIERCE) in the Institute of Geosciences at Goethe University Frankfurt. Different

185 carbonate phases were targeted. The system features a 193 nm RESOlution M-50 laser ablation (LA)

186 system with two-volume LaurinTechnic ablation cell connected to a ThermoScientific Element XR

187 sector field inductively coupled plasma mass spectrometer (ICP-MS). All analyses were performed as

188 'spot' measurements, utilising relatively low repetition rates to ablate through the thin sections

189 analysed here. Ablation took place with He in the outer cell (400 ml/min) and Ar sample gas from the

190 ICP-MS (1.025 l/min) mixed into the lower-volume inner cell. $N_2$ was admixed downstream of the

191 ablation cell (4.5 ml/min) to improve sensitivity (e.g. Durrant, 1994). The ICP-MS was fitted with a Ni

192 jet sample cone and Ni H skimmer cone and operated in medium resolution mode. Tuning of the

193 system was performed to maximise sensitivity (~6 M cps $^{238}U$ in low resolution mode; NIST SRM612

194 60 μm, 6 Hz, 6 J/cm$^2$) while maintaining the oxide and doubly charged production rate below 0.5% and

195 2% respectively. Monitored masses included $^{11}B$, $^{23}Na$, $^{24}Mg$, $^{25}Mg$, $^{27}Al$, $^{39}K$, $^{55}Mn$, $^{56}Fe$, $^{88}Sr$, $^{138}Ba$, $^{208}Pb$,

196 and $^{238}U$.

197 Samples were ablated using a 50 μm diameter laser beam at 4 Hz with an on-sample fluence of 5

198 J/cm$^2$. Quantification was performed using $^{43}Ca$ as the internal standard and NIST SRM610 as the

199 external standard. Data processing was performed using an in-house Matlab script following

200 established procedures (Heinrich et al., 2003), described in detail elsewhere (Evans and Müller, 2018).

201 Briefly: all analyses were baseline corrected by subtracting the mean of the two adjacent gas-blank

202 datasets, normalised to $^{43}Ca$, and standardised into element/Ca molar ratios using the analyte/$^{43}Ca$

203 count/concentration ratio from NIST SRM610. No significant drift in this count ratio was observed for

204 any relevant analyte for the session reported here such that standardisation was based on the mean

205 of all available NIST analyses. Down-hole elemental fractionation relative to Ca was corrected by

206 calculating least-squares 3$^{rd}$-order polynomials through the NIST SRM610 element/$^{43}Ca$-depth data,

207 which were then used as the sweep-time specific ratios for sample quantification. The NIST SRM610

208 values of Jochum et al. (2011) were used in all cases except Mg, for which we use that of Pearce et al.

209 (1997) following the data analysis of Evans and Müller (2018).

210 Data quality was assessed via repeat analysis of the 'nanopellet' version of the CaCO$_3$ standards JCp-1

211 and MACS-3 (Garbe-Schönberg and Müller, 2014; Jochum et al., 2019), ablated in an identical manner

212 to the samples. Pooling together all analyses on both standards (n = 20) and comparing to the reported

213 values of Jochum et al. (2019) yields an accuracy of <5% for $^{11}B$, $^{56}Fe$, $^{88}Sr$, <10% for $^{23}Na$, $^{24+25}Mg$, $^{138}Ba$,

214 $^{238}U$, 25% for $^{27}Al$, 30% for $^{39}K$, and 70% in the case of $^{55}Mn$, comparable to the achievable long-term

215 data quality from a similar system (Evans and Müller, 2018, albeit using a quadrupole ICP-MS).

216 Precision (repeatability), defined as the 2SD of all individual analyses normalised to the reported value

was better than 10% in the case of [11]B, [23]Na, [88]Sr, [238]U, <15% for [24+25]Mg, [39]K, [56]Fe, [138]Ba, 45% for [27]Al,
and 130% for [55]Mn. We stress that all of these values are strongly concentration-dependent and may
under/overestimate sample data quality depending on the analyte concentration of a given sample
analysis.
*2.6 Inductively Coupled Plasma Optical Emission Spectrometry (ICP-OES)*
Minor element analyses for Mg, Sr, Na, Mn, Fe, S, P, Al and Rb were undertaken on selected
subsamples of dried, powdered, bulk glendonites using an Agilent 5110 VDV ICP-OES at the Camborne
School of Mines, University of Exeter, following the method detailed in Ullmann et al. (2020). The
minor element data are expressed as ratios to Ca and calibrated using certified single-element
standards mixed to match the chemical composition of the analysed samples. Precision and accuracy
of the analyses were measured and controlled by interspersing multiple measurements of
international reference materials, JLs-1 and quality control solutions (BCQ2 and BCQ3). Analytical
uncertainty of element/Ca ratios in these reference materials is less than 1 % (2 RSD) at concentrations
> 100 times the quantification limit (measured as 6 SD of the variability of blank solutions in a batch
run, n = 5). For lower concentrations, the uncertainty of individual measurements is similar to the
quantification limit, i.e. 6 μmol/mol for Mg/Ca, 0.2 μmol/mol for Sr/Ca, 0.02 mmol/mol for Na/Ca, 5
μmol/mol for Mn/Ca, 5 μmol/mol for Fe/Ca, 0.2 mmol/mol for S/Ca, 0.08 mmol/mol for P/Ca, 0.06
mmol/mol for Al/Ca, and 0.04 mmol/mol for Rb/Ca.
*2.7 Microprobe data, IODP Exp. 396 ashes*
A representative selection (n=13) of Exp. 396 ashes from Holes U1567A, U1567C, U1568A, U1568B,
U1569A, and U1570D were analysed for their major element composition. Individual silicate glass
grains were picked and mounted in epoxy for matrix glass analysis. Polished and carbon coated grain
mounts were analysed on a Cameca SX100 electron microprobe with 5 wavelength dispersive
spectrometers (WDS) at University of Oslo. Analyses were conducted with an accelerating voltage of
15 kV and a 10 nA beam current using a defocused beam size of 10 μm. Counting times were 10 s for
Na, Si, Cl, K, S, Fe, Al, Mg, and Mn; and 20 for Ca and Ti.
*2.8 PHREEQC modelling*
The PHREEQC model (version 3; Parkhurst and Appelo, 2013, using the phreeqc.dat database) was
used to calculate pore water chemical speciation and saturation indices for Hole U1568A, based on
multiple shipboard analyses of interstitial water (Planke et al., 2023a,b). The model input values were
those from the ICP-AES for consistency. The *in situ* pressures (in atm; 1 atm = 101 kPa) were estimated
using the measured water depth of 1707.4 m below surface (mbsf) and a pressure gradient of 0.1
atm/m in the water column. In the subsurface, a pressure gradient of 0.2 atm/m was assumed, giving
a pressure of 209 atm at the deepest interstitial water sample depth (192.4 m). The PHREEQC model
was run as a batch reactor for each solution using a table input into SOLUTION_SPREAD, then a second
simulation RUN_CELLS to recalibrate the pe and pH from the ionic balance of the shipboard
measurements of cation and anion contents (Planke et al., 2023b,c).
**3. Results**
*3.1 Sedimentological context*
The glendonites discovered by IODP Expedition 396 are found within two holes of the Modgunn
transect, U1567C (3°3.219'E 65°21.785'N) and U1568A (3°3.109'E 65°21.594'N); and within two holes
of the Mimir transect, U1569A (2°1.608'E 65°49.878'N) and U1570A (1°59.623'E 65°49.890'N) (Fig. 2
and Table 1). For the Modgunn transect, the glendonites are all found within the latest Paleocene and
PETM-aged hydrothermal vent infilling deposits (Unit IV, Fig. 2), in horizons with the most numerous
and thickest ash intervals (Fig. 2). Lithological Unit IV is described as "Dark greenish grey to very dark
grey claystone to siltstone, with common volcanic ash beds and light bioturbation" (Planke et al.,
2023b, p.10). The glendonites in the Mimir cores are found in early Eocene-aged parts of the
succession, in Unit Va and Vb of Hole U1570A, also in close association with numerous and thick ash
layers. In Hole U1569A the glendonites are found at the very top of Unit Va, some 17 m above the
thickest and most numerous ash horizons, although notably in an interval with limited core recovery
(Fig. 2), which implies that these sediments may be extremely rich with unconsolidated ash horizons.
The deposits of Lithological Unit Va are described as "very dark grey mudstone with sparse parallel
lamination and slight bioturbation, with rare limestone intervals and common ash beds"; and those of
Vb as "very dark grey mudstone with sparse parallel lamination and slight bioturbation, with common
ash beds and diagenetic pyrite." (Planke et al., 2023c, p10 Sites U1569 & U1570). High-resolution logs
(5-cm scale) of the glendonite-bearing core sections and photographs of the *in situ* glendonites can be
found in the Supplementary Material.
*3.2 Organic carbon isotopes*
Organic carbon isotope data for the Mimir High (Holes U1570D, U1570A and U1569A) are shown on
Fig 2A. The carbon isotope data for the Modgunn hydrothermal vent complex is shown in Fig. 2B,
which includes new data from Hole U1568A and published data from Holes U1568B, U1567B, and
U1567C (Berndt et al., 2023). In the Modgunn boreholes, the PETM CIE is manifest as a negative CIE
of over 5 ‰ in Holes U1568A, U1567B, and U1567C, followed by a relaxation of $\delta^{13}C$ values back to c.
-1 ‰ compared to pre-PETM conditions for the rest of the vent infill. Hole U1568B currently has too
low data resolution, and Hole U1567A had extremely poor core recovery across the vent interval (Fig.
2B). The unusual PETM CIE shape at these localities is attributed to a change in the dominant carbon
source from terrestrial to marine-dominated organic matter into the extreme excursion and back to
terrestrially dominated organic matter for the minor excursion (Berndt et al., 2023). A similar CIE
shape has been observed nearby in the Grane field in the northern North Sea (Jones et al., 2019),
suggesting comparable paleoenvironmental conditions across the central part of the Nordic Seas area
at this time. The Mimir $\delta^{13}C_{org}$ data spans the PETM interval in the three cores, identified by the first
and last occurrences of *A. augustum* (Planke et al., 2023c). There is only a slight (1-2 ‰) $\delta^{13}C_{org}$
excursion is present in these holes, and it is difficult to differentiate between PETM and pre/post-
PETM strata on carbon isotopic evidence alone. The marine organic matter-rich CIE (~5 ‰) observed
at Modgunn and Grane has not been identified at Mimir from this dataset, but for Hole U1569A this
may be because there was poor core recovery over the critical PETM onset interval and the most
negative part of the CIE may be missing (Fig. 2A).
*3.3 Glendonite morphology*
The glendonites are variable in size and appearance, with some being cemented or partially cemented
(Fig. 3C, E, H and I), and some present as an uncemented amalgam of smaller crystals (Fig. 3B, D, F, G),
and some retaining their structure but as a porous mesh of calcite (Fig. 3A and J). In size, they range
from small fragments (2 – 5 mm across, Fig. 3A,D,F,I,J) to crystals up to or beyond the than the entire
width of the core (Fig. 3B,C) In some cases, the crystal appears to have grown over and incorporated
parts of the host sediment (Fig. 3E,H), yet in others appears to have either displaced the sediment it
was growing in (Fig. 3G), or grew up into the water column with later sedimentation burying it (Fig.
3C). Nearly all the recorded glendonite specimens can be ascribed to a single rosette morphotype (Fig.
3) (following terminology proposed by Frank et al., 2008), except for specimens from Fig. 3D, F, I and
J, in which the morphology is unclear due to the fragmented nature or disturbance of the structure
during drilling. They are generally pale beige to brown in colour, although cemented examples show
distinct areas within a cut 'crystal' of grey or white-pale brown carbonate (Fig. 3C,H). Some samples
are only small fragments of glendonite, identified by their characteristic bladed shape (e.g. Fig. 3A,E)
and/or the open, porous texture of the carbonate (e.g. Fig. 3F, I, J). Most of the glendonites are found
*in situ*, with the exception of sample U1567C-10X-3 40-45, where glendonite fragments were
identified entrained in the drill mud (Fig. 3A). Therefore, these fragments could have originated from
some short distance away from their final position within the core, likely the large glendonite found
in the core below (sample U1567C-11X-1 83-93; Fig. 3B).

*3.4 PXRD*

The PXRD analysis of the bulk composition of 5 different glendonites reveal that the glendonites are
mostly composed of calcite with a major to minor magnesian calcite component. Minor amounts of
quartz, halite, rhodochrosite ($MnCO_3$) and gypsum were also identified (Table 1 and Supplementary
Material Fig. S11).

*3.5 Microscopy*

Polished thin sections were examined by light microscopy and revealed several different carbonate
phases (Figs. 4 and 5). These have been grouped by appearance (colour, texture, relationship to other
phases) and geochemistry, and are described in detail in Table 2. Where possible, these phase names
are in keeping with previous studies' description of ikaite carbonate phases (e.g. Huggett et al., 2005;
Grasby et al., 2016; Vickers et al., 2018; Mikhailova et al., 2021; Scheller et al., 2021; Schultz et al.,
2023a; Counts et al., *accepted*)., with the main phase type assigned based on colour and sub-types
based on microstructure and geochemical data.

*3.6 LA ICP-MS*

The LA ICP-MS trace element data shows that Mg/Ca ratios are distinct on average between the
carbonate phases, with Type 1A ≤ 1B ≤ 2A = 2B ≤ 0, albeit with substantial heterogeneity within a
phase such that the ranges overlap in most cases (Fig. 6). There is also substantial overlap between
types 1 and 2 for all other measured elemental ratios (Fe/Ca, Mn/Ca, Sr/Ca, P/Ca, S/Ca; Fig. 6). The
measurements made from the outer hard crust of the glendonite sample from core U1569A-19R-2
show the highest Mg/Ca contents of all, and distinctly higher S/Ca than all other carbonate phases.
Type 0 has significantly higher Mn/Ca and P/Ca than all other measured calcite phases, and
significantly lower S/Ca than other measured carbonate phases (Fig. 6).

*3.7 ICP-OES*

Bulk drill-sampled ICP-OES results find that the Exp. 396 glendonites have Mg/Ca ratios in the range
of c. 20 - 50 mmol/mol, comparable to the Fur Formation carbonates (Fig 7). Sr/Ca range from 1.5 -
1.8 mmol/mol; Fe/Ca from 1.6 - 38 mmol/mol, with an outlier of 165 mmol/mol (1568A 15X 4 55-58);
Mn/Ca ratios range from 0.4 - 8.4; P/Ca ratios range from 1.5 - 12.8 mmol/mol; and S/Ca ratios range
from 0.8 - 5.5 mmol/mol (Fig. 7).

*3.8 Microprobe data, 396 ashes*

The Exp. 396 ashes all have a basaltic tholeiitic composition, similar to the Fur positive series ashes
(Fig. 8). However, they have relatively high MgO content ranging from 4.3 to 9.6 wt%, which is
generally higher than that of the Fur positive series (3.3-7.1 wt%; Larsen et al., 2003; Stokke et al.,
2020b).

*3.9 IW analysis and PHREEQC modelling*

The shipboard analyses of IW samples showed non-typical behaviour within Units IV and V (the vent
infill) at Site 1568A, with lower alkalinity (2–3 mM) and higher pH (~8.2) compared to both underlying
and overlying strata (Planke et al., 2023b). Many IW profiles in ocean sediments above igneous
basement show a marked reduction in dissolved Mg/Ca ratios with depth (e.g. Sites U1403-1405;
Norris et al., 2014), and two of the four holes where IW profiles were conducted at Modgunn and
Mimir (U1567A, U1569A). At Holes U1568A and U1570A, the two with observed glendonites within
the cored strata, the Mg/Ca profiles show an inversion to increasing ratios with depth between around
60 – 100 mbsf (Fig. 9C).
Results from the PHREEQC modelling for the IW are shown in Fig. 9 and in the supplementary data.
Dissolved inorganic carbon (DIC) speciation is shifted towards $[CO_3^{2-}]$ across the HTV infill and as a
result the saturation index of the $CaCO_3$ minerals increases in these horizons, in the case of calcite to
values greater than zero. This indicates that while the measured IW samples reflect a complex
diagenetic history, the zones with confirmed glendonite occurrences also have anomalous IW
chemistries and carbonate saturation states in present day pore waters.

**4. Discussion**

*4.1 Ikaite formation and transformation to glendonite*

The general microfabric of the glendonites is similar to that observed in previous work on glendonite
and transformed ikaite. Notably, areas of calcite blebs (bubble-like mineral inclusions) with Type 1A
cores with 2A overgrowths are also observed in glendonites from Cretaceous to Recent in age, and in
modern transformed ikaites (Huggett et al., 2005; Grasby et al., 2016; Vickers et al., 2018; Mikhailova
et al., 2021; Scheller et al., 2021; Schultz et al., 2023). The presence of a hardened carbonate rim is
also a common feature of glendonites (Fig. 4) (Grasby et al., 2017; Scheller et al., 2022; Schultz et al.,
2023a; Counts et al., 2023). However, the green Type 0 calcite identified in this study has not been
observed in other glendonite thin sections, and appears to be both visually and chemically distinct
from the other calcite phases measured in the Exp. 396 glendonites (Fig. 6).
Based on the geochemical data, we suggest the following model for the sequential formation of the
various calcite phases, as illustrated in Fig. 10:
Stage 1: The ikaite grew in the sediments, at or just below the sediment-water interface. The Type 0
phase appears to have grown directly onto/against the surface of an ikaite crystal, prior to its
decomposition, suggesting it may represent the earliest preserved phase present (Fig. 10).
Stage 2: Chemical and thermal conditions changed as burial continued, and the ikaite started to break
down (Fig. 10). The formation of 1A blebs began during the recrystallisation reaction, preferentially
excluding Mg from the crystal structure, leading to a highly localised increase of $Mg^{2+}$ in the pore
waters. This may have taken place over timescales of years (Schultz et al., 2023b). Where breakdown
was rapid, larger areas of Type 1B formed (also observed in ikaite transformed on timescales of hours;
Schultz et al., 2023b). The faster mineralisation rate of 1A compared to 1B is believed to have caused
less discrimination against Mg in the crystal structure, as evidenced by the higher Mg content of 1B
compared to 1A (Fig.6). Then, Types 2A and 2B grew, incorporating more Mg into the calcite structure
than the Types 1 due to its now higher concentration in the local waters.
Stage 3: An undetermined length of time later, after continued burial, Type 3 fibrous syntaxial and/or
isopachous sparry calcite forms in some of the pore spaces in the ikaite, growing from the surface of
Types 1 and/or 2 calcite phases (Fig. 10). Note that all phases grouped as Types 1 and 2 are believed
to have formed directly from ikaite (i.e. the $CaCO_3$ is believed to have been entirely or dominantly
sourced directly from the ikaite), whereas we propose that the Type 3 calcite formed from a later
diagenetic fluid (e.g. Vickers et al., 2020; Counts et al., 2023). This is based on microfabric and
geochemical (elemental and isotopic) data that suggests a very different source fluid. There is very
little Type 3 calcite in the Exp. 396 glendonites; much less than is observed in those of the Fur
Formation (Vickers et al., 2020). The variable proportion of Type 3 calcite in glendonite is likely
responsible for the observed higher bulk Mg/Ca, Mn/Ca and P/Ca ratios than in transformed modern
ikaite (Fig. 7).
*4.2 Relationship to North Atlantic volcanism and timing of ikaite formation*
The glendonites of the Exp. 396 cores are found in latest Paleocene, PETM, and (post-PETM) early
Eocene-age sediments, within or just above the intervals containing the most numerous and thickest

ash layers (Fig. 2 and 11). The only other reported glendonites from this time interval (Paleocene and early Eocene) are also found in the Nordic Seas region, close to the eruptive sites of the North Atlantic Igneous Province (NAIP). Glendonites have been identified in both Paleocene and Eocene strata in Svalbard (Spielhagen and Tripati, 2009; Cui et al., 2021), and in early Eocene strata in northern Denmark (Huggett et al., 2005; Vickers et al., 2020; Fig. 1). In the Danish Paleogene successions, glendonites are only found within the early Eocene Fur Formation that contains over 140 macroscopic ash layers of dominantly tholeiitic basalt composition (Fig. 11; Stokke et al., 2020b; Vickers et al., 2020). The Fur Formation corresponds to the ash-rich Balder and Tare formations offshore in the North Sea and Norwegian margin, respectively (King, 2016). The close stratigraphic association with ash layers within the Nordic Basins suggests that ash deposition and/or diagenesis may play a critical role in ikaite precipitation, in contrast to ikaites found in most modern settings. The breakdown of sedimentary $C_{org}$ via sulphate reduction, and/or the anaerobic oxidation of methane (AOM) are thought to play a key role in ikaite precipitation, largely because the low $\delta^{13}C$ values measured in ikaites and glendonites suggest an organic or methanogenic source of carbon (e.g. Rogov et al., 2023 and references therein), and also because these organic matter decomposition processes generate DIC (Hiruta and Matsumoto, 2022; Whiticar et al. 2022). Methane has been linked to ikaite/glendonite formation due to their frequent proximity to methane seeps (Greinert and Derkachev, 2004; Teichert and Luppold, 2013; Hiruta and Matsumoto, 2022) and gas inclusions containing methane and other hydrocarbons in glendonite specimens from the Jurassic of Siberia (Morales et al., 2017). However, other studies which examined sedimentary biomarker evidence for AOM in Oligocene and Eocene-aged glendonite-bearing strata did not find evidence for significantly elevated rates of AOM and support an organic matter source for the examined glendonites (Qu et al., 2017; Vickers et al., 2020).

For ikaite to be precipitated over the more stable $CaCO_3$ polymorphs, factors inhibiting calcite and promoting ikaite precipitation are also required. These may include high alkalinity, high concentrations of phosphate and/or $Mg^{2+}$, and low temperatures (Rickaby et al., 2006; Zhou et al., 2015; Purgstaller et al., 2017; Stockmann et al., 2018).  The NAIP may have played a key role in generating just such conditions for $CaCO_3$ precipitation and the precipitated polymorph being ikaite. Hydrothermal venting of methane and other gases occurred at sites proximal to the NAIP (Svensen et al., 2004; Frieling et al., 2016; Jones et al., 2019; Berndt et al., 2023); and the explosive nature of the NAIP eruptive volcanism could have driven short-term climate cooling ('volcanic winters', e.g. Robock et al., 2000; Schmidt et al., 2016; Stokke et al., 2020a). Furthermore, the large amounts of volcanic ash in the sediments likely underwent rapid diagenesis, generating the chemical conditions in the pore waters that could have inhibited calcite precipitation and therefore promoted ikaite precipitation (e.g. Gislason and Oelkers, 2011; Olsson et al., 2014; Murray et al., 2018).  Indeed, ikaite and other

carbonates were discovered as travertine in the Hvanná river in the vicinity of the Eyjafjallajökull
volcano shortly after eruptive activity began in Spring of 2010 (Olsson et al., 2014).
Compositional data from Exp. 396 shows that the ashes, like those from the Fur positive series (Stokke
et al., 2020b), are tholeiitic (Fig. 8). Basaltic volcanic material undergoes rapid chemical weathering
(e.g. Gislason and Oelkers, 2011), thus we suggest that the chemical alteration of these tholeiitic ashes
could have generated the chemical conditions which promoted ikaite formation, which may include
increasing alkalinity and $[Ca^{2+}]$, driving changes in aqueous Mg/Ca (e.g. Gislason and Oelkers, 2011;
Olsson et al., 2014; Purgstaller et al., 2017; Murray et al., 2018; Tollefsen et al., 2020). The interstitial
water measurements from the cores reflect the conditions in the core interstitial waters today, some
c. 55 million years since deposition, and therefore do not reflect conditions immediately prior to ikaite
precipitation. Nonetheless, these pore water profiles may provide remnant signatures of post-
depositional processes and therefore shed light on how ash diagenesis may have altered local pore
water chemistry, promoting ikaite precipitation.
The pore water profiles show a strong, sharp change in pH and carbon speciation across the ash-
bearing intervals (Fig. 11; Planke et al., 2023b,c), likely a retained signal of the dissolution/leaching of
the ashes themselves. While pore water chemistry certainly continued to evolve over the last 55
million years, the signature of ash diagenesis on early Paleogene pore waters may have been
effectively retained, being buffered against later fluid flow and associated later overprinting reactions
by the over- and under-lying clay-rich strata (Planke et al., 2023b,c; Fig. 2). High smectite contents
formed from the weathering of silicate material such as basaltic ash both *in situ* and in terrestrial
catchment areas, can result in 'aquitard' deposits characterised by very low permeability and low
effective porosity (Hendry et al., 2015). This allows large concentration gradients to develop within
the sediment pile, as documented along ash-rich margins globally (Torres et al. 1995 and references
therein). The major fluctuations in pH, alkalinity, and Mg/Ca ratios across these ash-rich intervals at
Modgunn and Mimir (Planke et al., 2023b,c) suggests that these pore waters are likely to have evolved
in semi-isolation from over- and underlying strata. Therefore, if the pore water system is closed, the
evolution of the pore water chemistry may be limited. Alternatively, long-term diagenesis of ash
minerals may be continually altering pore water chemistry across these intervals with some diffusive
exchange across the low-permeability clay boundaries, yet still providing an indication of the degree
to which ash alteration may impact fluid composition. Pore water measurements through selected
cores show that the glendonite-bearing levels associated with volcanic ash are coincident with
relatively elevated pH and lower alkalinity, although alkalinity change across ash-rich intervals is less
abrupt (Fig. 11). The carbonate chemistry of these pore waters is controlled by a complex balance
between $CaCO_3$ dissolution/precipitation, respiration of organic carbon, and at these sites, probably
by alkalinity generation from the dissolution of ash (Longman et al. 2021). As such, it is difficult to
ascribe the observed pore water carbonate chemistry changes (Fig. 11) to any one process. However,
the relatively high pH and low alkalinity relative to the over- and underlying mud-rich sediments is
consistent with alkalinity generation via ash dissolution, resulting in increased TAlk and pH, being
counteracted by $CaCO_3$ precipitation, drawing down carbonate alkalinity. Our observation of abundant
$CaCO_3$ in the form of ikaite and later diagenetic phases in these ash-rich sediments is consistent with
this hypothesis. Further, PHREEQC simulations indicate a clear difference in carbonate speciation
across ash-rich, glendonite bearing intervals with lower $[CO_2]_{aq}$, and higher $[CO_3^{2-}]$ and $[CaCO_3^0]$ (Fig.
9) compared with surrounding intervals. Aragonite and calcite saturation indices go from under- to
oversaturated, and dolomite shows enhanced oversaturation, across these intervals (Fig. 9). Overall,
while this indicates that the conditions necessary for ikaite formation prevailed in the hydrothermal
vent infill (Fig. 9), we note that the carbonate chemistry of the remnant pore water profiles is
characterised by conditions that are far less saturated with respect to $CaCO_3$ minerals than those
necessary to form ikaite in the laboratory, especially at higher temperatures (Purgstaller et al., 2017;
Tollefsen et al., 2020).
*4.3 Timing and conceptual model for ikaite growth*
Zhou et al. (2015) examined modern marine sedimentary ikaite bearing sites and identified "Ikaite
Formation Zones" (IFZs) based on where the decreasing $[Ca^{2+}]$ downcore profile intersects the
increasing DIC profile (calculated by maximum $[Ca^{2+}]$ x DIC). The IFZ is thus highly variable, as it
depends on the amount of organic matter (carbon source) present, which in turn is linked to processes
such as sedimentation rate and primary productivity. In modern settings, this is generally between 2
– 15 m within the sediment pile (Zhou et al., 2015). In the early Paleogene Norwegian Margin, infill
sedimentation rates of hydrothermal vent complex craters were very high. Taking the very
conservative estimate of a hydrothermal vent infill duration of 43 kyr for the Modgunn crater (Berndt
et al., 2023), the glendonite-bearing horizons for the vent infill cores would have been at depths of
below 15 m within 8 kyr for Hole U1568A and 16.5 kyr for Hole U1567C. For sediments such as those
encountered in Modgunn and Mimir, the primary driver of DIC was likely AOM, since there was
significant methane venting around the NAIP at the time (Svensen et al., 2004; Frieling et al., 2016);
and TOC in the sediments is relatively low (generally < 1.5 wt %; Planke et al., 2023b,c), which suggests
that bacterial sulphate reduction of organic matter could not have been the main source of carbon
.The source of $[Ca^{2+}]$ is believed to result from diagenesis of the ashes (e.g. Gislason and Oelkers,
2011). Given that the diagenesis of fresh volcanic material occurs very rapidly after deposition (on

timescales of shorter than a month; Hembury et al., 2012; Olsson et al., 2014), the ikaite formation zones are likely to have been much shallower than 15 m at the time of ikaite precipitation, possibly near the sediment-water interface. Observations on the relationship between the Exp. 396 glendonites and the host sediments support this theory of very shallow ikaite formation depths. For example, it can be seen that the pale grey sediment is displaced immediately around the glendonite shown in Fig. 3G, yet an ash layer 2 cm above shows no displacement. In Fig. 3H, the pale grey sediment that the glendonite has grown in is displaced, yet the ash that is deposited on top, two thirds up the height of the glendonite, shows no displacement at the top, suggesting the top of the ikaite crystal was potentially protruding into the water column from the seafloor at the time of the ash fall. Therefore, it can be deduced that these parent ikaites grew in the soft sediment prior to compaction, possibly just centimetres below the sediment-water interface. In the Danish succession, similar "boudinage" textures are observed around both glendonites and tree branch fossils in the Fur Formation (Schultz et al., 2022). This indicates that the ikaites grew at or close to the sediment-water interface, as they exhibit the same relationship to the host sediment as tree branches that came to rest on top of or sticking into the top few centimetres or decimetres of sediment. In case of the Exp. 396 glendonites, which occur in rapidly deposited sediments, the parent ikaites must have grown quickly, on timescales of years, for these relationships to be observed. Modern ikaites have been known to grow to cm-scale crystals on timescales of months to years (Boch et al., 2015; Schultz et al., 2022, 2023a), suggesting that this is indeed possible. In one case, one month after final construction of a concrete riverbed in Alpine Austria, cm-thick ikaite crystal aggregates were discovered to have formed (Boch et al., 2015); in another instance, ikaite crystals 3 cm long were found growing next to a reservoir causeway in Utqiaġvik, Alaska, some 3 years after its construction (Schultz et al., 2022; 2023a).

Thus, we argue that the parent ikaites to Exp. 396 glendonites grew on geologically synchronous timescales to the sediments in which they are found. Their occurrence and rapid growth during the Paleocene-Eocene greenhouse climate were likely facilitated by the unique conditions near the active NAIP. We propose a scenario in which the chemical environment that stabilised ikaite was provided by the NAIP hydrothermal vent formation and early and rapid diagenesis of the (frequent) ashes deposited from this LIP. The explosive NAIP volcanism may have also may also have played a critical role in driving short-term (c.  sub-decadal scale) cooling which additionally helped promote the formation of ikaite.

Paleotemperature estimates for the PETM and early Eocene from both biomarkers and stable and clumped isotope thermometry for the North Atlantic and Nordic Seas region (e.g. (Schoon et al., 2015;

Stokke et al., 2020; Vickers et al., 2020; Meckler et al., 2022; Rush et al., 2023) show variable temperatures and intervals of cooling which are apparently at odds with global records (e.g. Westerhold et al., 2020)). This has led to speculation that short, sharp, transient cooling events occurred, possibly only on a regional scale (Stokke et al., 2020a; Vickers et al., 2020), as evidence for sudden/short duration cooling is absent from some nearby and other northern Hemisphere sites from similar time periods (e.g. Belgian/Paris basin, Evans et al., 2018; West Siberian Sea and the Arctic, Frieling et al., 2014 and references therein), all of which were characterised by SST of ~20-30°C during this time. Alternatively, however, these discrepancies may reflect changes in sourcing, production depth or seasonal biases in the specific proxies used (e.g. Jia et al., 2017; Udoh et al., 2022; de Winter et al., 2021), or, in the case of carbonate clumped isotopes, solid-state reordering pushing the apparent temperature up (Henkes et al., 2014), or kinetic disequilibium during the carbonate precipitation (Daëron et al., 2019), meaning that these signals may be challenging to interpret as reflecting true aquatic temperatures at the time of formation in some settings.

Regional-scale cooling episodes may result from, for example, a series of short-lived volcanic 'winters' caused by sulphur degassing during effusive and explosive eruptions (Robock et al., 2000; Schmidt et al., 2016; Stokke et al., 2020a). Given the sub-decadal residence time of sulphur in the atmosphere, this climate forcing would require many closely-spaced eruptions to maintain its cooling effect on timescales of centuries to millennia (Jones et al., 2016). However, since ikaite appears to have grown very rapidly in some of the successions in the Nordic Seas, a growth rate of this magnitude does not seem implausible.

In addition to possible volcanically-driven regional climatic events, the unique paleogeography of the region were likely critical for providing the conditions required for ikaite formation. The Nordic Seas were a series of hydrographically and/or geographically semi-restricted, relatively shallow basins with varying connectivity to the Atlantic, Tethys, and Arctic oceans across the Paleocene–Eocene interval (Fig. 1), with several lines of evidence suggesting that maximum restriction occurred in the post-PETM early Eocene (Zacke et al., 2009; Hovikoski et al., 2021; Jones et al., 2023). Such restricted conditions could imply bottom waters in individual basins (simultaneously or at different times) may have been effectively isolated from the global ocean. Without sufficient exchange with the global ocean, bottom waters colder than the global deep ocean, that may have formed during transient cool conditions such as anomalously long and cold (e.g., volcanic-driven) winters, could have led to the formation of a cold bottom layer in a heavily stratified water column in the Nordic Sea basins for a prolonged period. Finally, the thermal uplift and emplacement of a continental flood basalt province created an extensive high-altitude plateau to the west of the shallow seaways that marked the region, which is

likely to have altered the positioning, oscillation, and intensity of the Paleogene northern hemisphere
jet stream (Jones et al., 2023). While no studies have investigated this potential effect, it is reasonable
to assume that a regional microclimate was likely present in the Nordic Seas region at this time. It is
possible that a combination of factors led to a bias towards the recording of winter temperatures in
the bottom waters of the basin (Vickers et al., 2020). If a long, severe cooling forcing such as a
sustained volcanic winter occurred, cooling of the surface waters could eventually have triggered a
dense water cascade, bringing these cold, dense waters to the bottom of the basin system. Being
restricted and stratified, these cold waters could potentially have remained at the bottom of the basin
system for a long time (years) (e.g. Vickers et al., 2020) such that, together with ample volcanic ash
supply, it seems plausible that the local conditions coincided to create the necessary conditions for
ikaite growth in a hothouse climate.
**5. Conclusions**
Glendonites are found throughout the Late Paleocene and early Eocene sediments from the IODP Exp.
396 cores, including those deposited during the PETM, and closely associated with the volcanic ashes
from the nearby NAIP. High-resolution examination and sedimentary logging of these cores reveals
ten glendonite horizons, six in the post-PETM sections recovered from the Mimir transect, three from
the PETM interval of the hydrothermal vent infill collected in the Modgunn Transect, and one from
the pre-PETM interval of this hydrothermal vent infill. Observations of their relationship to their host
sediments suggest they grew within centimetres of the sediment-water interface, sometimes even
protruding into the water column. Based on known time-scales of ash diagenesis and ikaite growth,
we argue that the parent ikaites grew rapidly, within timescales of years to decades after ash
deposition. Examination of thin-sections of the glendonites via a number of geochemical methods
reveals that the Exp. 396 glendonites show a range of carbonate phases, including fabrics not
previously observed in other glendonites nor transformed natural ikaite. These features suggest that
their parent ikaite growth environment was unusual even for ikaite, and the leaching and rapid early
diagenesis of the NAIP volcanic ashes likely generated the required pore water conditions that
stabilised ikaite over other calcium carbonate polymorphs. Paleothermometry studies for the Nordic
Seas Region during this time suggest that seawater temperatures were punctuated by remarkably
cold, short-term events, although no such temperature deviations have been found outside of this
region. Glendonites are also found in the Paleocene and Eocene succession of Svalbard, and in the
early Eocene (post-PETM) succession of Denmark, but nowhere outside of this semi-enclosed shallow
basin.

The close association of glendonites to ash in the Exp. 396 succession, and likewise in the early Eocene of Denmark, along with biomarker and clumped isotope thermometry evidence of episodic transient cooling events, supports our theory that the eruptive phases of the NAIP led to ikaite precipitation in the Nordic Seas region. This was both a chemical and thermal effect: the early diagenesis of the ashes likely drove pore water conditions chemically favourable to ikaite, and the eruptions could have caused transient volcanic winters that were much colder than the prevailing background climate of the time. The unique paleogeography of the region may have led to the basin bottom waters being biased towards these anomalously cold temperatures for years or even decades, allowing the ikaites to grow to the centimetre to >decimetre sizes we observe. Further work is required to test this hypothesis, including detailed, high resolution multi-proxy temperature reconstructions e.g. via clumped isotope thermometry of the glendonites themselves, and biomarker and palynological assemblage-based temperature reconstructions for the sediments of these cores.

**Figures and Tables**

**Figure 1: (A)** Paleogeographic map of the Nordic Seas region with North Atlantic Igneous Province volcanism shown, after Jones et al. (2023). Location of all known Paleocene – Eocene glendonite bearing sites marked – Exp. 396 Modgunn and Mimir cores (this study); Paleogene-Eocene sediments of Svalbard (Spielhagen and Tripati, 2009), and early Eocene Fur Formation of northern Denmark (Vickers et al., 2020). **(B)** High-resolution 3-D seismic data for holes 1568 and 1567 along the Modgunn transect (from Planke et al., 2023b). **(C)** High-resolution 3-D seismic data for holes 1569 and 1570 along the Mimir Transect (from Planke et al., 2023c). Holes from which glendonites were recovered are shown in red. PETM intervals are shown in yellow.

**Figure 2:** Overview logs of the cores with glendonite horizons marked by the red glendonite cartoon. **(A)** The Mimir (U1569 - U1570) transect, from Planke et al. (2023c). The PETM interval (pale yellow) is identified by biostratigraphy (Planke et al., 2023c) and carbon isotope stratigraphy (this study). **(B)** The Modgunn (U1567 - U1568) transect, from Planke et al. (2023b) and Berndt et al., 2023. Core sections of PETM age are highlighted in yellow, and the hydrothermal vent infill (e.g. Fig. 1B) is shown in grey. All correlations between cores are supported by lithologic change, biostratigraphic zonation (Planke et al., 2023b, Berndt et al., 2023), and carbon isotope stratigraphy (Berndt et al., 2023; this study).

**Figure 3:** Photographs of glendonites *in situ* in the cores from the Modgunn and Mimir transects. **(A)** Glendonite fragments in drill mud from 1567C-10X-3 40-45 (MLV 86). **(B)** Glendonite from section 1567C-11X-1 94-95 (MLV 57, 97). **(C)** Cemented glendonite from section 1569A-19R-2 54-62 (MLV 90).

**(D)** Porous carbonate mush interpreted as glendonite from section 1568A-15X-4 (MLV 88). **(E)** Porous
cemented glendonite incorporating host sediment from section 1568A-15X-1. **(F)** Glendonite from
section 1570A-15R-1 108-112 (MLV 92). **(G)** Glendonite fragment in 1570A-15R-1 22-25 (MLV 91). **(H)**
Glendonite from section 1570A-25R-1 (MLV 93). (I) Small cemented glendonite fragment from section
1570A-22R-2. **(J)** *In situ* fragment of glendonite from section 1570A-24R-1.
**Figure 4:** Photomicrographs of polished thin sections from selected Exp. 396 glendonites. The blue
background colour is derived from the resin rather than the glendonite. **(A)** and **(B)** show the typical
harder outer rim with more porous centre characteristic of transformed ikaite (e.g. Schultz et al.,
2023a). Red dots labelled 8, 9 and 10 are spots where LA ICP-MS analysis was performed. The
glendonites commonly show areas of different calcite types defined by colour, which are often hard
to place into the "traditional" carbonate phase types seen in other glendonites (e.g. Huggett et al.,
2005; Vickers et al., 2018). **(C)** shows a distinct boundary between white Type 2B calcite and brown
Type 1B calcite, neither of which show zoning defined by colour or porosity. **(D)** shows the sharp
boundary between green Type 0 carbonate, with black dendritic surface growth, and other calcite
phases. The shape of the sharp boundary that Type 0 defines on one side suggests that Type 0 grew
on the surface of and out from an ikaite crystal, which later broke down to leave void space and
patches of Type 1B with 2B overgrowths. **(E)** and **(F)** show patches of more typical zoned calcite blebs,
here labelled 1A and 2A, which appear to fit into the traditional categories of "Type I" (zoned brown
calcite forming the centre of the blebs) and "Type II" (zoned pale overgrowths on Type I; e.g. Vickers
et al., 2018; Schultz et al., 2023a). **(G)** Apparent reversal of the "typical" glendonite fabric, whereby
the central area of the calcite blebs is pale/white Type 2B and the overgrowth brown Type 1B calcite.
This contrasts with **(H)** which shows dark Type 1A with white Type 2A overgrowths.
**Figure 5:** Light microscopy, SEM photomicrographs and EDS element maps from thin sections of
glendonites at 1569A-19R-2 and 1567C-11X-1. **(A)** Overview under plane polarised light of the area
examined for glendonite at 1569A-19R-2, with the carbonate phases labelled. **(B)** BSE image of the
same area. Higher porosity in the Type 1 can be seen. **(C)** BSE image of zoomed in area of Type 2B with
Type 1B overgrowth. Higher porosity of Type 1 is again clear. **(D)** EDS map showing Mg distribution
across calcite types 1B and 2B, overlaid on the BSE photomicrograph. **(E)** The same map without the
BSE photomicrograph **(F)** EDS map showing Mg distribution across calcite types 1A, 2A and 2B. **(G)**
Overview under plane polarised light of the area examined for glendonite at 1567C-11X-1. **(H)** BSE
image of the same area, with pop-out **(I)** showing the microcrystalline nature of Type 0. **(J)**
Magnification of the same area with types IB and 2B calcite under BSE. **(K)** EDS element map showing
the Mg distribution across the same area.

**Figure 6:** LA ICP-MS element/Ca data for points across the Exp. 396 glendonite polished thin sections. The data have been grouped according to the calcite types described in the main text and in the preceding figures. Photomicrograph showing the location points 8 – 11 from outer edge inwards are shown bottom right, and also in Fig. 4A. Photomicrographs showing the location of all the individual points measured may be found in the Supplementary Material.

**Figure 7:** Element/Ca ratios of the Exp. 396 glendonites and associated calcites compared to published ICP-OES data for other glendonite-bearing sites.

**Figure 8: (A)** A Total Alkali Silica (TAS) plot comparing the Exp. 396 ashes (this study) to published data for both positive (Stokke et al. 2020b) and negative (Larsen et al., 2003) ash series of the Fur Formation in northern Denmark. The Exp. 396 ashes and Fur positive series fall into the basaltic fields, whereas the Fur negative series show much more variation and have overall more felsic compositions. Note that while the Fur positive series data are microprobe analyses of matrix glass, the Fur Negative series data are whole rock data. However, the whole rock samples were leached of clay prior to analysis and no significant dilution is expected. **(B)** Ternary Alkali-Iron-Magnesium (AFM) diagram showing that the basaltic ashes from both the Exp. 396 sites and the Fur positive series are tholeiitic basalts. Note that many of the Exp. 396 ashes have higher MgO content than the Fur positive ashes.

**Figure 9:** (A) PHREEQC simulation results for carbonate speciation in the U1568A core, which spans the hydrothermal vent infill (grey highlight labelled 'HTV'). Note that $HCO_3^-$ (the major species) is not shown. (B) PHREEQC simulation saturation indices for several carbonate polymorphs. (C) IW Mg/Ca profiles for all glendonite-bearing cores (Planke et al., 2023b,c).

**Figure 10:** Schematic of ikaite transformation in the Exp. 396 cores, adapted from Counts et al. (2023) based on observed textural relationships and geochemistry of the calcite phases in the Exp. 396 glendonites.

**Figure 11: (A)** Relative position of glendonites in the Paleocene-Eocene sediments of selected cores from the mid-Norwegian Margin, Exp. 396, compared to measured ash thicknesses. Pore water alkalinity and pH data (Planke et al., 2023b,c) are also shown. Pale grey indicates the PETM-aged intervals in the stratigraphy. Note that for U1569A, core recovery was poor, particularly in the bottom, ash-bearing part (see Fig. 2). High ash contents lead to lower core recovery as they are course-grained and unlithified; therefore it is likely that there were much more numerous and thicker ash horizons in the interval between 18R and 37R (c. 180 – 340 mbsf). **(B)** Relative position of glendonites in the Paleocene-Eocene sediments of Northern Denmark, compared to ash thicknesses per metre (Jones et

al., 2023). Glendonite horizons for the Fur Formation are from Vickers et al., (2020) (solid lines) and dashed line as identified by Henrik Friis, pers. comm. Pale grey indicates the end of the body of the PETM carbon isotope excursion (Jones et al., 2023). The recovery phase is between ashes -33 and -21a. SC = Stolleklint Clay.

**Table 1:** Glendonites of the Exp. 396 cores, PXRD data from bulk glendonite analysis, element/Ca ratios.

**Table 2:** Descriptions of the different carbonate phases observed within the glendonites through thin section microscopic and geochemical analysis (light microscopy, SEM, EDS and LA-ICP-MS).

**Team List**

The IODP expedition 396 Scientists are: S. Planke, C. Berndt, C.A. Alvarez Zarikian, A. Agarwal, G.D.M. Andrews, P. Betlem, J. Bhattacharya, H. Brinkhuis, S. Chatterjee, M. Christopoulou, V.J. Clementi, E.C. Ferré, I.Y. Filina, J. Frieling, P. Guo, D.T. Harper, M.T. Jones, S. Lambart, J. Longman, J.M. Millett, G. Mohn, R. Nakaoka, R.P. Scherer, C. Tegner, N. Varela, M. Wang, W. Xu, and S.L. Yager.

**Author contribution**

MLV designed the study, undertook sampling and high-resolution logging, photographed and examined thin-sections under a light microscope and SEM, co-ran LA-ICPMS analysis, lead the writing of the manuscript.

MTJ undertook PHREEQC modelling, assisted in the interpretation of the data and co-wrote the manuscript.

JL undertook the ash thickness measurements, assisted in the interpretation of the data, co-wrote the manuscript and assisted with the logging and sampling.

DE ran the LA-ICPMS and assisted in the interpretation of the data and writing of the manuscript.

CVU undertook the ICP-OES analysis.

ES carried out microprobe analysis of the ashes and plotting of this data.

MV undertook the PXRD analysis.

JF assisted in the interpretation of the data and co-wrote the manuscript.
DTH assisted in the interpretation of the data and co-wrote the manuscript.
VJC assisted in the interpretation of the pore-water data and co-wrote the manuscript.
IODP E396 S: undertook drilling of the cores, all shipboard analysis, and sampling.
**Competing interests**
The authors declare that they have no conflict of interest.
**Acknowledgements**
This research used samples and/or data provided by the International Ocean Discovery Program
(IODP). IODP is funded by the US National Science Foundation (NSF), Japan's Ministry of Education,
Culture, Sports, Science and Technology (MEXT), The European Consortium for Ocean Research
Drilling (ECORD), China's Ministry of Science and Technology (MOST), Australian-New Zealand IODP
Consortium (ANZIC), and India's Ministry of Earth Science (MoES). We thank the master, crew, and
technical support staff of the JOIDES Resolution during IODP Expedition 396. We gratefully
acknowledge funding for this study from the European Commission, Horizon 2020 (ICECAP; grant no.
101024218 to MLV) and from the Research Council of Norway through the Centres of Excellence
funding scheme, project numbers 223272 (CEED), and 332523 (PHAB), and the Goldschmidt
Laboratory national infrastructure (project number 295894). JF acknowledges funding from UK IODP
grant NE/W007142/1. We personally thank Ray Leadbitter and Independent Petrographic Services Ltd
for making the thin sections used in this study, Siri Simonsen for use and running of the SEM at the
University of Oslo. VJC was supported by NSF grant OCE-2205921. FIERCE is financially supported by
the Wilhelm and Else Heraeus Foundation and by the Deutsche Forschungsgemeinschaft (DFG: INST
161/921-1 FUGG, INST 161/923-1 FUGG and INST 161/1073-1 FUGG), which is gratefully
acknowledged. This is FIERCE contribution No. 136. We thank Dr. John W. Counts for his input on
discussions of ikaite-glendonite transformation. We thank our reviewers, Dr. Mikhail Rogov, Dr. Niels
de Winter and one anonymous reviewer for their insightful constructive comments and feedback.

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

Methods, in Planke, S., Berndt, C., Alvarez Zarikian, C.A., and the Expedition 396 Scientists, Mid-
Norwegian Margin Magmatism and Paleoclimate Implications. Proceedings of the International Ocean
Discovery Program, 396. https://doi.org/10.14379/iodp.proc.396.102.2023, 2023d.Purgstaller, B.,
Dietzel, M., Baldermann, A. and Mavromatis, V. Control of temperature and aqueous Mg2+/Ca2+ ratio
on the (trans-) formation of ikaite. Geochim. Cosmochim. Ac., 217, 128-143.
https://doi.org/10.1016/j.gca.2017.08.016, 2017.
Qu, Y., Teichert, B.M.A., Birgel, D., Goedert, J.L. and Peckmann, J. The prominent role of bacterial
sulfate reduction in the formation of glendonite: a case study from Paleogene marine strata of western
Washington State. Facies, 63, 1-16. https://doi.org/10.1007/s10347-017-0492-1, 2017.Rickaby, R.,
Shaw, S., Bennitt, G., Kennedy, H., Zabel, M. and Lennie, A. Potential of ikaite to record the evolution
of oceanic δ18O. Geology, 34(6), 497-500. https://doi.org/10.1130/G22413.1, 2006.
Robock, A. Volcanic eruptions and climate. Rev. Geophys., 38(2), 191-219.
https://doi.org/10.1029/1998RG000054, 2000.
Rogov, M.A., Ershova, V.B., Shchepetova, E.V., Zakharov, V.A., Pokrovsky, B.G., and Khudoley, A.K.
Earliest Cretaceous (late Berriasian) glendonites from Northeast Siberia revise the timing of initiation
of transient Early Cretaceous cooling in the high latitudes: Cretaceous Res., 71, 102–112.
https://doi.org/10.1016/j.cretres.2016.11.011, 2017.
Rogov, M., Ershova, V., Vereshchagin, O., Vasileva, K., Mikhailova, K. and Krylov, A. Database of global
glendonite and ikaite records throughout the Phanerozoic. Earth Syst. Sci. Data, 13(2), 343-356.
https://doi.org/10.5194/essd-13-343-2021, 2021.
Rogov, M., Ershova, V., Gaina, C., Vereshchagin, O., Vasileva, K., Mikhailova, K. and Krylov, A.
Glendonites throughout the Phanerozoic. Earth-Sci. Rev., 104430.
https://doi.org/10.1016/j.earscirev.2023.104430, 2023.
Röhl, U., Westerhold, T., Bralower, T.J. and Zachos, J.C. On the duration of the Paleocene-Eocene
thermal maximum (PETM). Geochemistry, Geophysics, Geosystems, 8(12).
https://doi.org/10.1029/2007GC001784, 2007.
Rush, W., Self-Trail, J., Zhang, Y., Sluijs, A., Brinkhuis, H., Zachos, J., Ogg, J.G. and Robinson, M.
Assessing environmental change associated with early Eocene hyperthermals in the Atlantic Coastal
Plain, USA. Clim. Past, 19, 1677-1698. https://doi.org/10.5194/cp-19-1677-2023y, 2023.
Scheller, E.L., Grotzinger, J. and Ingalls, M. Guttulatic calcite: A carbonate microtexture that reveals
frigid formation conditions. Geology, 50(1), 48-53. https://doi.org/10.1130/G49312.1, 2022.
Schmidt, A., Skeffington, R.A., Thordarson, T., Self, S., Forster, P.M., Rap, A., Ridgwell, A., Fowler, D.,
Wilson, M., Mann, G.W. and Wignall, P.B. Selective environmental stress from sulphur emitted by
continental flood basalt eruptions. Nat. Geosci., 9(1), 77-82. https://doi.org/10.1038/ngeo2588, 2016.
Schoon, P.L., Heilmann-Clausen, C., Schultz, B.P., Damasté, J.S.S., Schouten, S. Warming and
environmental changes in the eastern North Sea Basin during the Palaeocene–Eocene Thermal
Maximum as revealed by biomarker lipids. Org. Geochem. 78, 79-88.
https://doi.org/10.1016/j.orggeochem.2014.11.003, 2015.
Schultz, B., Thibault, N. and Huggett, J. The minerals ikaite and its pseudomorph glendonite: Historical
perspective and legacies of Douglas Shearman and Alec K. Smith. P. Geologist. Assoc., 133(2), 176-192.
https://doi.org/10.1016/j.pgeola.2022.02.003, 2022.
Schultz, B.P., Huggett, J.M., Kennedy, G.L., Burger, P., Friis, H., Jensen, A.M., Kanstrup, M., Bernasconi,
S.M., Thibault, N., Ullmann, C.V., and Vickers, M.L. Petrography and geochemical analysis of Arctic
ikaite pseudomorphs from Utqiaġvik (Barrow), Alaska, Norw. J. Geol., 103, 202303.
https://dx.doi.org/10.17850/njg103-1-3, 2023a.
Schultz, B.P., Huggett, J., Ullmann, C.V., Kassens, H. and Kölling, M. Links between Ikaite Morphology,
Recrystallised Ikaite Petrography and Glendonite Pseudomorphs Determined from Polar and Deep-
Sea Ikaite. Minerals, 13(7), 841. https://doi.org/10.3390/min13070841, 2023bSluijs, A., Frieling, J.,
Inglis, G.N., Nierop, K.G., Peterse, F., Sangiorgi, F. and Schouten, S. Late Paleocene–early Eocene Arctic
Ocean sea surface temperatures: reassessing biomarker paleothermometry at Lomonosov Ridge.
Clim. Past, 16(6), 2381-2400. https://doi.org/10.5194/cp-16-2381-2020, 2020.
Spielhagen, R.F. and Tripati, A. Evidence from Svalbard for near-freezing temperatures and climate
oscillations in the Arctic during the Paleocene and Eocene. Palaeogeogr. Palaeocl., 278(1-4), 48-56.
https://doi.org/10.1016/j.palaeo.2009.04.012, 2009.
Stockmann, G.J., Seaman, P., Balic-Zunic, T., Peternell, M., Sturkell, E., Liljebladh, B. and Gyllencreutz,
R. Mineral Changes to the Tufa Columns of Ikka Fjord, SW Greenland. Minerals, 12(11), 1430.
https://doi.org/10.3390/min12111430, 2022.
Stockmann, G., Tollefsen, E., Skelton, A., Brüchert, V., Balic-Zunic, T., Langhof, J., Skogby, H. and
Karlsson, A. Control of a calcite inhibitor (phosphate) and temperature on ikaite precipitation in Ikka
Fjord, southwest Greenland. Appl. Geochem., 89, 11-22.
https://doi.org/10.1016/j.apgeochem.2017.11.005, 2018.
Stokke, E.W., Jones, M.T., Tierney, J.E., Svensen, H.H. and Whiteside, J.H. Temperature changes across
the Paleocene-Eocene Thermal Maximum–a new high-resolution TEX86 temperature record from the
Eastern North Sea Basin. Earth Planet. Sc. Lett., 544, 116388.
https://doi.org/10.1016/j.epsl.2020.116388, 2020a.
Stokke, E.W., Liu, E.J., Jones, M.T. Evidence of explosive hydromagmatic eruptions during the
emplacement of the North Atlantic Igneous Province. Volcanica, 3 (2), 227-250.
https://doi.org/10.30909/vol.03.02.227250, 2020b.
Suess, E., Balzer, W., Hesse, K.F., Müller, P.J., Ungerer, C.T. and Wefer, G. Calcium carbonate
hexahydrate from organic-rich sediments of the Antarctic shelf: precursors of glendonites. Science,
216(4550), 1128-1131. https://doi.org/10.1126/science.216.4550.1128, 1982.
Svensen, H., Planke, S., Malthe-Sørenssen, A., Jamtveit, B., Myklebust, R., Rasmussen Eidem, T. and
Rey, S.S. Release of methane from a volcanic basin as a mechanism for initial Eocene global warming.
Nature, 429(6991), 542-545. https://doi.org/10.1038/nature02566, 2004.
Teichert, B.M.A. and Luppold, F.W. Glendonites from an Early Jurassic methane seep—Climate or
methane indicators?. Palaeogeography, Palaeoclimatology, Palaeoecology, 390, 81-93.
https://doi.org/10.1016/j.palaeo.2013.03.001, 2013.Tollefsen, E., Balic-Zunic, T., Mörth, C.M.,
Brüchert, V., Lee, C.C. and Skelton, A., 2020. Ikaite nucleation at 35 C challenges the use of glendonite
as a paleotemperature indicator. Sci Rep.-UK, 10(1), 8141. https://doi.org/10.1038/s41598-020-
996 64751-5, 2020.

Torres, M.E., Marsaglia, K.M., Martin, J.B. and Murray, R.W. Sediment diagenesis in western Pacific
basins. Geoph. Monog., 88, 241-258., 1995.
Ullmann, C.V., Boyle, R., Duarte, L.V., Hesselbo, S.P., Kasemann, S.A., Klein, T., Lenton, T.M., Piazza, V.
and Aberhan, M. Warm afterglow from the Toarcian Oceanic Anoxic Event drives the success of deep-
adapted brachiopods. Sci Rep.-UK, 10(1), 6549. https://doi.org/10.1038/s41598-020-63487-6, 2020.
Vickers, M., Watkinson, M., Price, G.D. and Jerrett, R., 2018. An improved model for the ikaite-
glendonite transformation: evidence from the Lower Cretaceous of Spitsbergen, Svalbard. Norw. J.
Geol., 98(1), 1 – 15 https://dx.doi.org/10.17850/njg98-1-01, 2018.
Vickers, M.L., Price, G.D., Jerrett, R.M., Sutton, P., Watkinson, M.P. and FitzPatrick, M., 2019. The
duration and magnitude of Cretaceous cool events: Evidence from the northern high latitudes. Geol.
Soc. Am. Bull., 131(11-12), 1979-1994. https://doi.org/10.1130/B35074.1, 2019.
Vickers, M.L., Lengger, S.K., Bernasconi, S.M., Thibault, N., Schultz, B.P., Fernandez, A., Ullmann, C.V.,
McCormack, P., Bjerrum, C.J., Rasmussen, J.A. and Hougård, I.W. Cold spells in the Nordic Seas during
the early Eocene Greenhouse. Nat. Comm., 11(1), 4713. https://doi.org/10.1038/s41467-020-18558-

1011    7, 2020.

Vickers, M.L., Vickers, M., Rickaby, R.E., Wu, H., Bernasconi, S.M., Ullmann, C.V., Bohrmann, G.,
Spielhagen, R.F., Kassens, H., Schultz, B.P. and Alwmark, C. The ikaite to calcite transformation:
Implications for palaeoclimate studies. Geochim. Cosmochim. Ac., 334, 201-216.
https://doi.org/10.1016/j.gca.2022.08.001, 2022.
Westerhold, T., Marwan, N., Drury, A.J., Liebrand, D., Agnini, C., Anagnostou, E., Barnet, J.S., Bohaty,
S.M., De Vleeschouwer, D., Florindo, F. and Frederichs, T. An astronomically dated record of Earth's
climate and its predictability over the last 66 million years. Science, 369(6509), 1383-1387.
https://doi.org/10.1126/science.aba6853, 2020.
Whiticar, M.J., Suess, E., Wefer, G. and Müller, P.J. Calcium carbonate hexahydrate (ikaite): History of
mineral formation as recorded by stable isotopes. Minerals, 12(12), 1627.
https://doi.org/10.3390/min12121627, 2022.
Zachos, J.C., Dickens, G.R. and Zeebe, R.E. An early Cenozoic perspective on greenhouse warming and
carbon-cycle dynamics. Nature, 451(7176), 279-283. https://doi.org/10.1038/nature06588, 2008.
Zacke, A., Voigt, S., Joachimski, M.M., Gale, A.S., Ward, D.J., Tütken, T. Surface-water freshening and
high-latitude river discharge in the Eocene North Sea. J. Geol. Soc. London, 166, 969-980.
https://doi.org/10.1144/0016-76492008-068, 2009.
Zhou, X., Lu, Z., Rickaby, R.E., Domack, E.W., Wellner, J.S. and Kennedy, H.A., 2015. Ikaite abundance
controlled by porewater phosphorus level: Potential links to dust and productivity. J. Geol., 123(3),
269-281. https://doi.org/10.1086/681918, 2015.

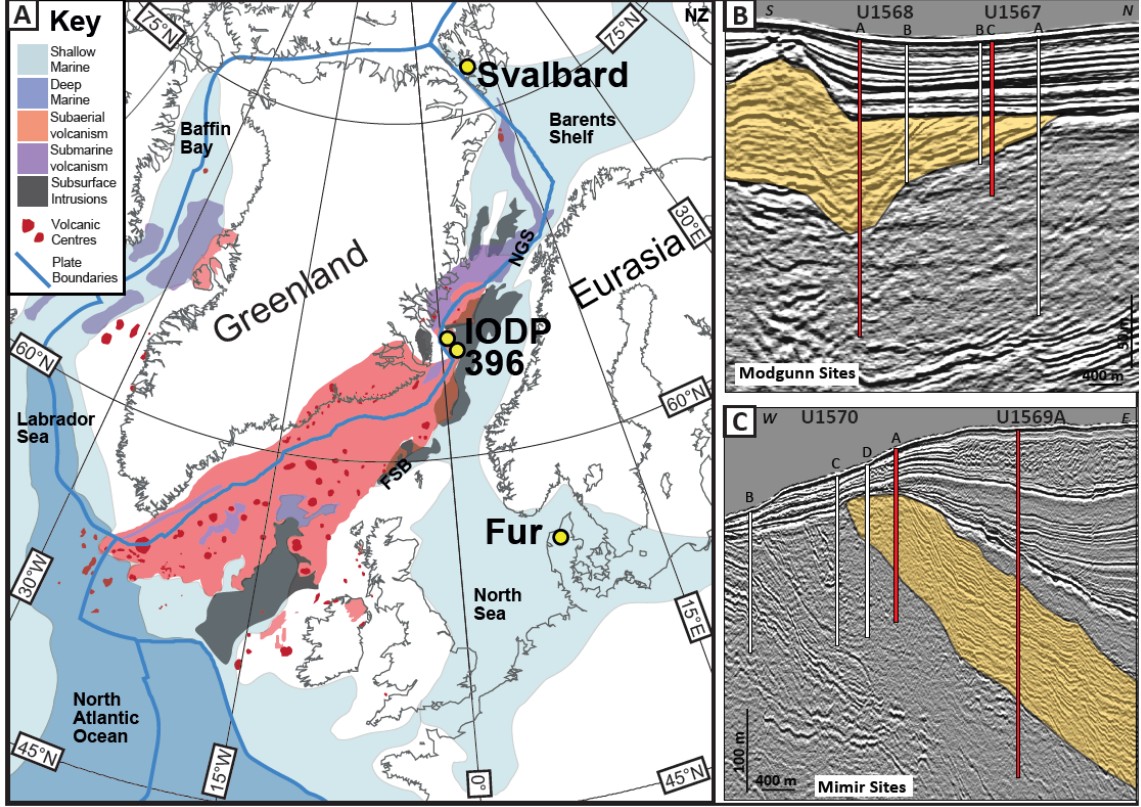

**Figure 1: (A)** Paleogeographic map of the Nordic Seas region with North Atlantic Igneous Province volcanism shown, after Jones et al. (2023). Location of all known Paleocene – Eocene glendonite bearing sites marked – Exp. 396 Modgunn and Mimir transects (this study); Paleogene-Eocene sediments of Svalbard (Spielhagen and Tripati, 2009), and early Eocene Fur Formation of northern Denmark (Vickers et al., 2020). **(B)** High-resolution 3-D seismic data for holes 1568 and 1567 along the Modgunn transect (from Planke et al., 2023). **(C)** High-resolution 3-D seismic data for holes 1569 and 1570 along the Mimir Transect (from Planke et al., 2023). Holes from which glendonites were recovered are shown in red. PETM intervals are shown in yellow.


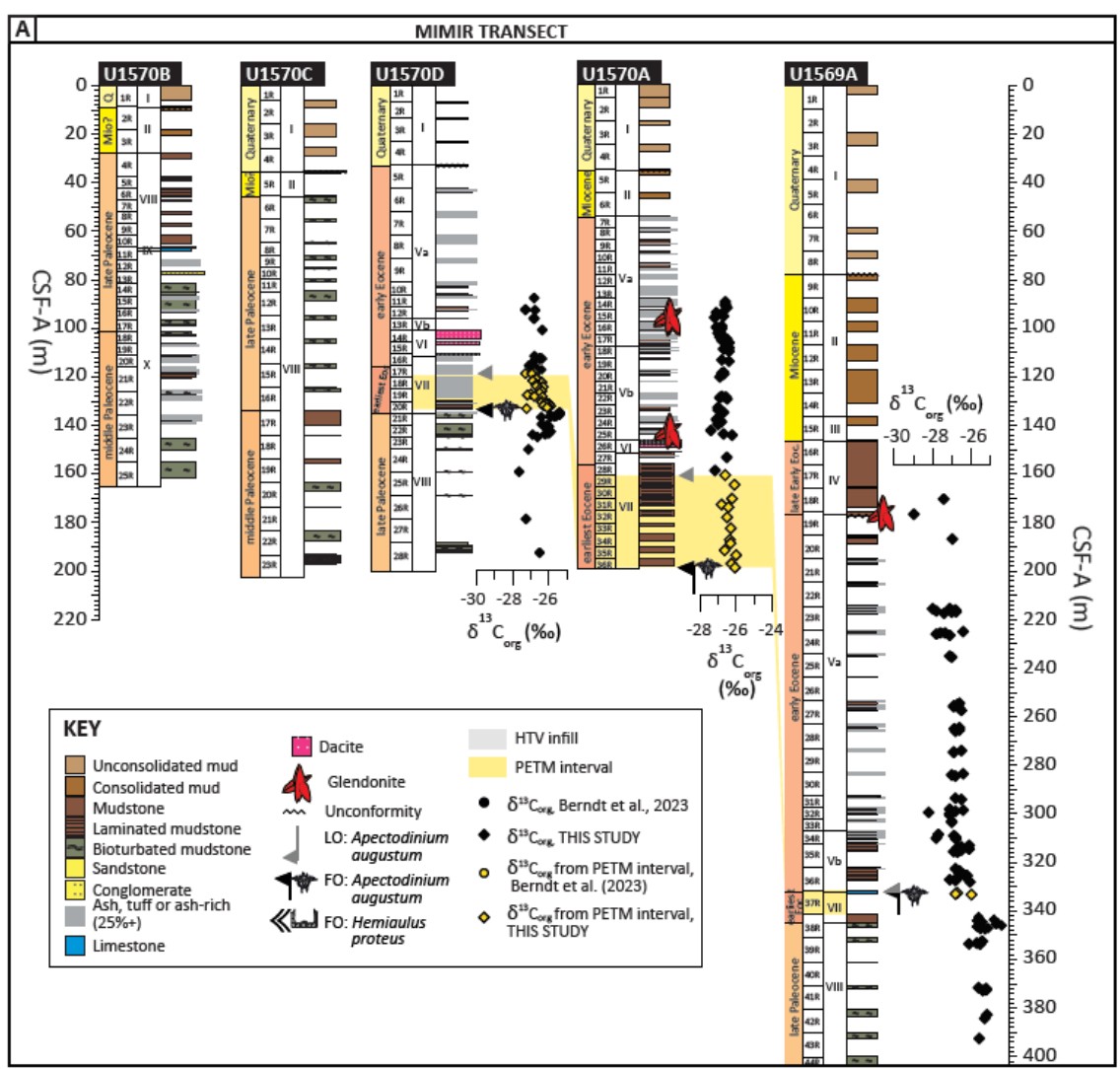

**Figure 2: Overview logs of the cores with glendonite horizons marked by the red glendonite cartoon.**
**(A)** The Mimir (U1569 - U1570) transect, from Planke et al. (2023c). The PETM interval (pale yellow) is identified by biostratigraphy (Planke et al., 2023c) and carbon isotope stratigraphy (this study). **(B)** The Modgunn (U1567 - U1568) transect, from Planke et al. (2023b) and Berndt et al., 2023. Core sections of PETM age are highlighted in yellow, and the hydrothermal vent infill (e.g. Fig. 1B) is shown in grey. All correlations between cores are supported by lithologic change, biostratigraphic zonation (Planke et al., 2023b, Berndt et al., 2023), and carbon isotope stratigraphy (Berndt et al., 2023; this study).


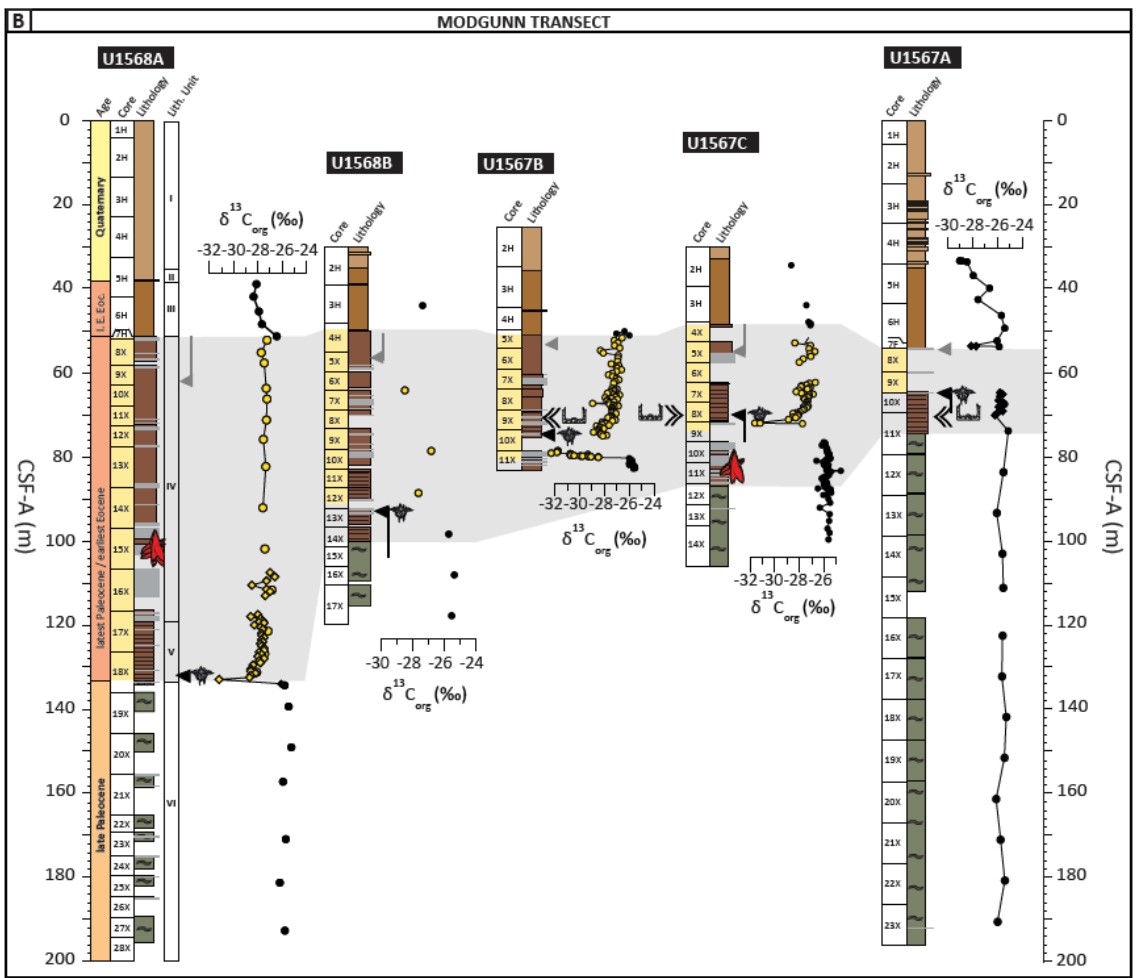

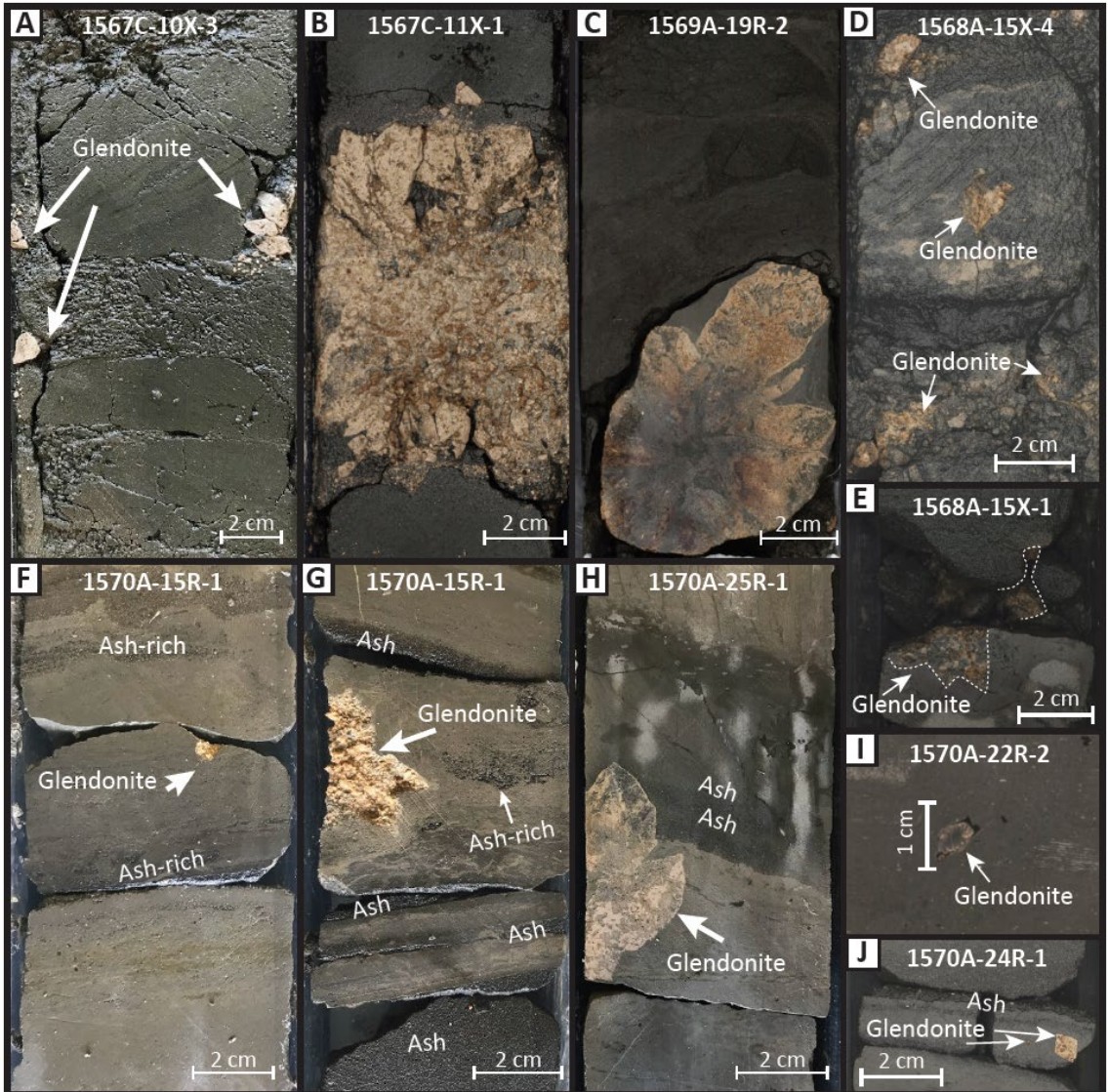

**Figure 3:** Photographs of glendonites *in situ* in the cores from the Modgunn and Mimir transects. **(A)** Glendonite fragments in drill mud from 1567C-10X-3 40-45 (MLV 86). **(B)** Glendonite from section 1567C-11X-1 94-95 (MLV 57, 97). **(C)** Cemented glendonite from section 1569A-19R-2 54-62 (MLV 90). **(D)** Porous carbonate mush interpreted as glendonite from section 1568A-15X-4 (MLV 88). **(E)** Porous cemented glendonite incorporating host sediment from section 1568A-15X-1. **(F)** Glendonite from section 1570A-15R-1 108-112 (MLV 92). **(G)** Glendonite fragment in 1570A-15R-1 22-25 (MLV 91). **(H)** Glendonite from section 1570A-25R-1 (MLV 93). (I) Small cemented glendonite fragment from section 1570A-22R-2. **(J)** *In situ* fragment of glendonite from section 1570A-24R-1.

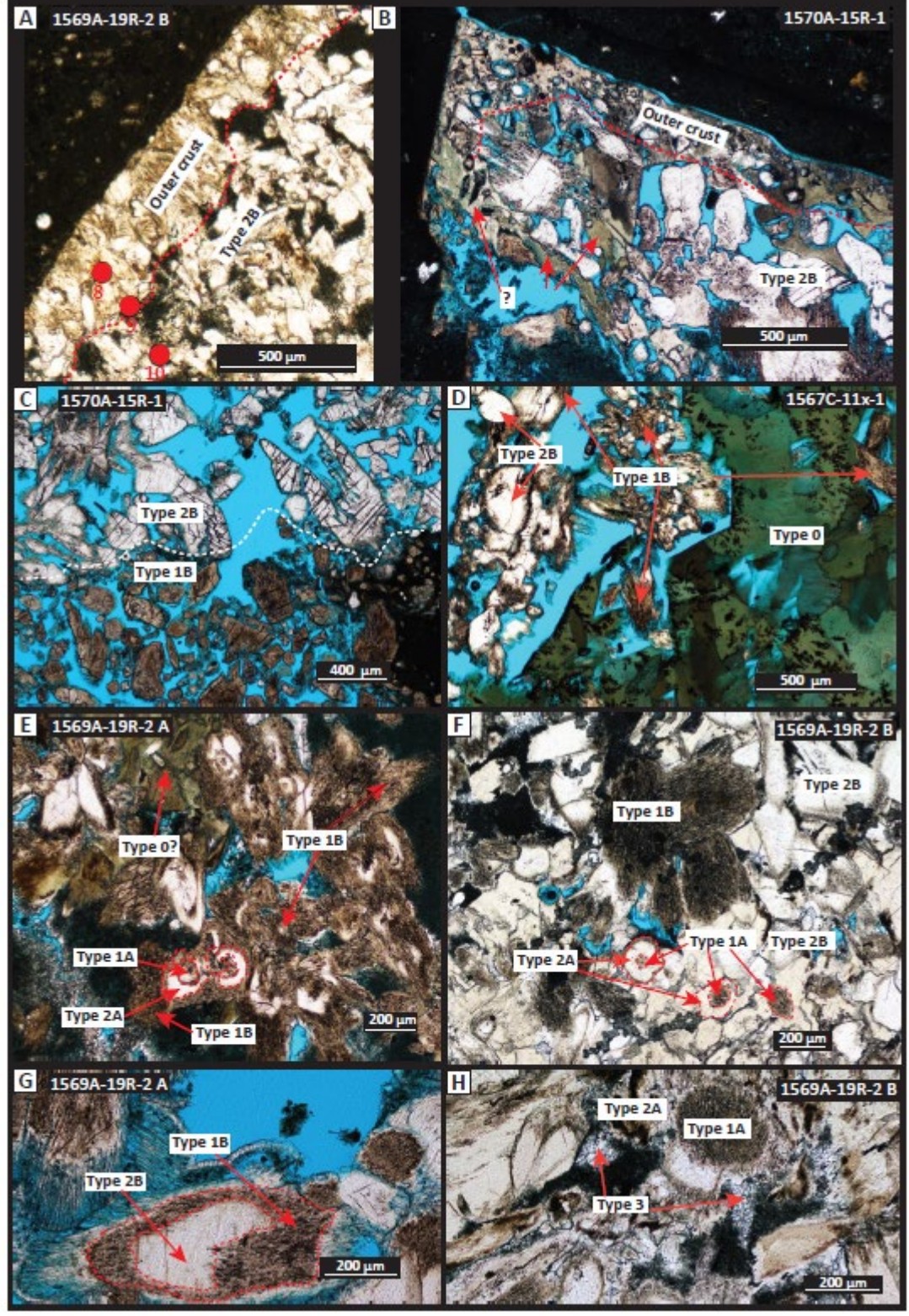

**Figure 4:** Photomicrographs of polished thin sections from selected Exp. 396 glendonites. The blue background colour is derived from the resin rather than the glendonite. **(A)** and **(B)** show the typical harder outer rim with more porous centre characteristic of transformed ikaite (e.g. Schultz et al., 2023). Red dots labelled 8, 9 and 10 are spots where LA ICP-MS analysis was performed. The glendonites commonly show areas of different calcite types defined by colour, which are often hard to place into the "traditional" carbonate phase types seen in other glendonites (e.g. Huggett et al., 2005; Vickers et al., 2018). **(C)** shows a distinct boundary between white Type 2B calcite and brown Type 1B calcite, neither of which show zoning defined by colour or porosity. **(D)** shows the sharp boundary between green Type 0 carbonate, with black dendritic surface growth, and other calcite phases. The shape of the sharp boundary that Type 0 defines on one side suggests that Type 0 grew on the surface of and out from an ikaite crystal, which later broke down to leave void space and patches of Type 1B with 2B overgrowths. **(E)** and **(F)** show patches of more typical zoned calcite blebs, here labelled 1A and 2A, which appear to fit into the traditional categories of "Type I" (zoned brown calcite forming the centre of the blebs) and "Type II" (zoned pale overgrowths on Type I; e.g. Vickers et al., 2018; Schultz et al., 2023). **(G)** Apparent reversal of the "typical" glendonite fabric, whereby the central area of the calcite blebs is pale/white Type 2B and the overgrowth brown Type 1B calcite. This contrasts with **(H)** which shows dark Type 1A with white Type 2A overgrowths.

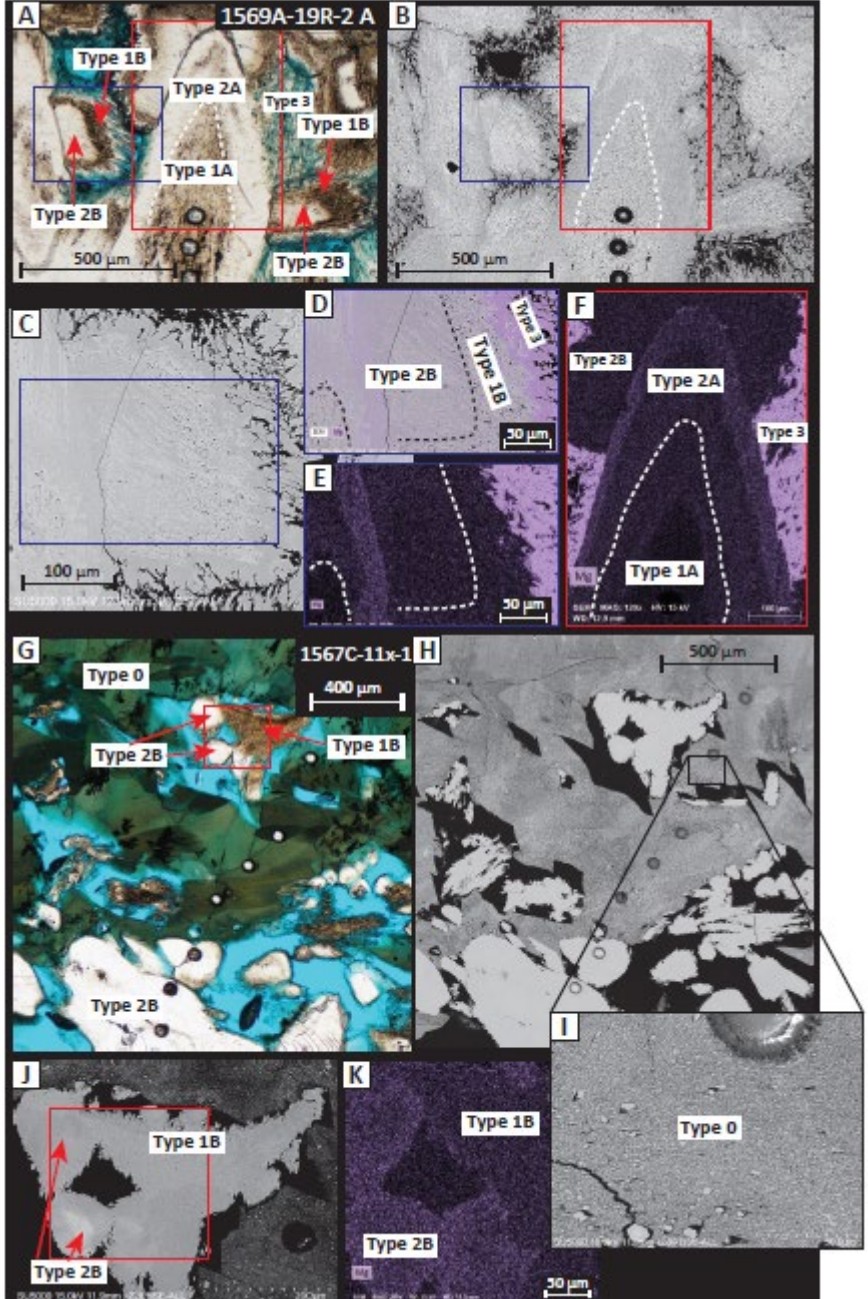

**Figure 5:** Light microscopy, SEM photomicrographs and EDS element maps from thin sections of glendonites at 1569A-19R-2 and 1567C-11X-1. **(A)** Overview under plane polarised light of the area examined for glendonite at 1569A-19R-2, with the carbonate phases labelled. **(B)** BSE image of the same area. Higher porosity in the Type 1 can be seen. **(C)** BSE image of zoomed in area of Type 2B with Type 1B overgrowth. Higher porosity of Type 1 is again clear. **(D)** EDS map showing Mg distribution across calcite types 1B and 2B, overlaid on the BSE photomicrograph. **(E)** The same map without the BSE photomicrograph **(F)** EDS map showing Mg distribution across calcite types 1A, 2A and 2B. **(G)** Overview under plane polarised light of the area examined for glendonite at 1567C-11X-1. **(H)** BSE image of the same area, with pop-out **(I)** showing the microcrystalline nature of Type 0. **(J)** Magnification of the same area with types IB and 2B calcite under BSE. **(K)** EDS element map showing the Mg distribution across the same area.

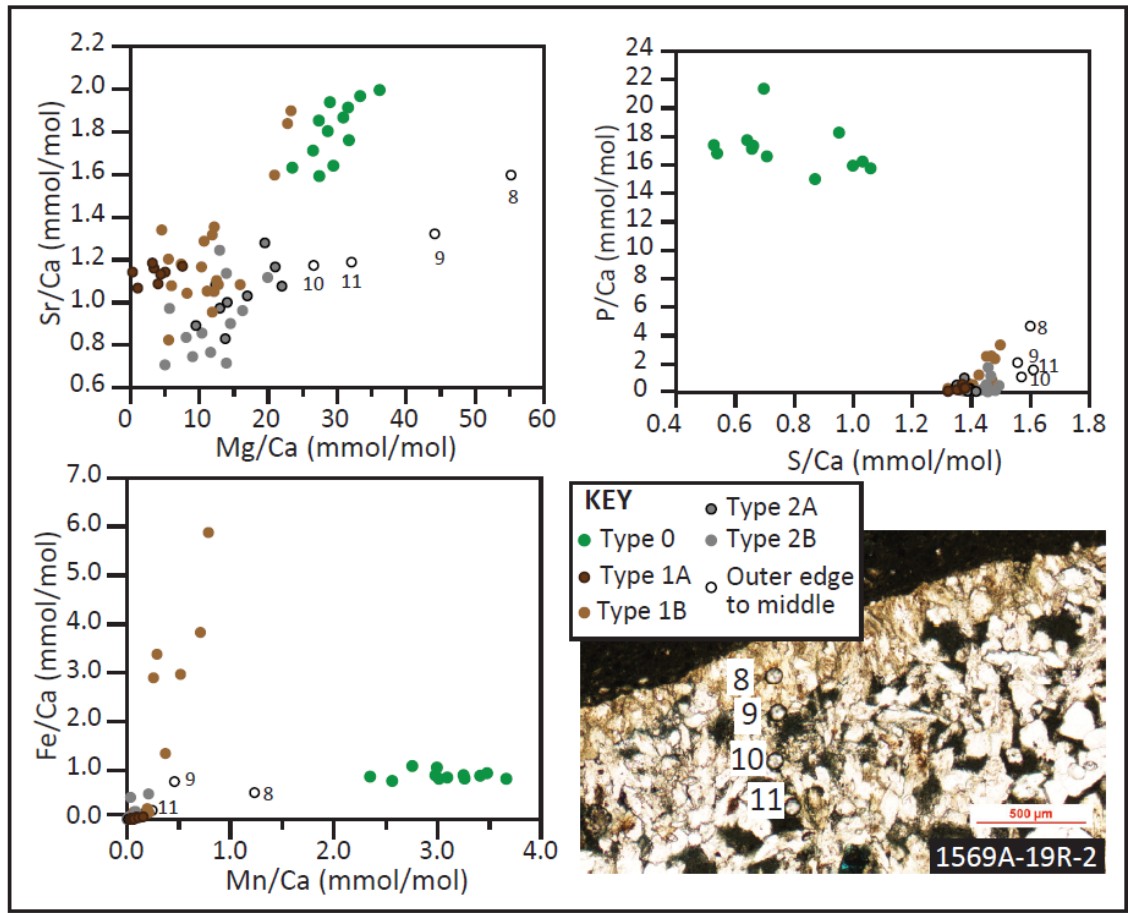

**Figure 6:** LA ICP-MS element/Ca data for points across the Exp. 396 glendonite polished thin sections. The data have been grouped according to the calcite types described in the main text and in the preceding figures. Photomicrograph showing the location points 8 – 11 from outer edge inwards are shown bottom right, and also in Fig. 4A. Photomicrographs showing the location of all the individual points measured may be found in the Supplementary Material.

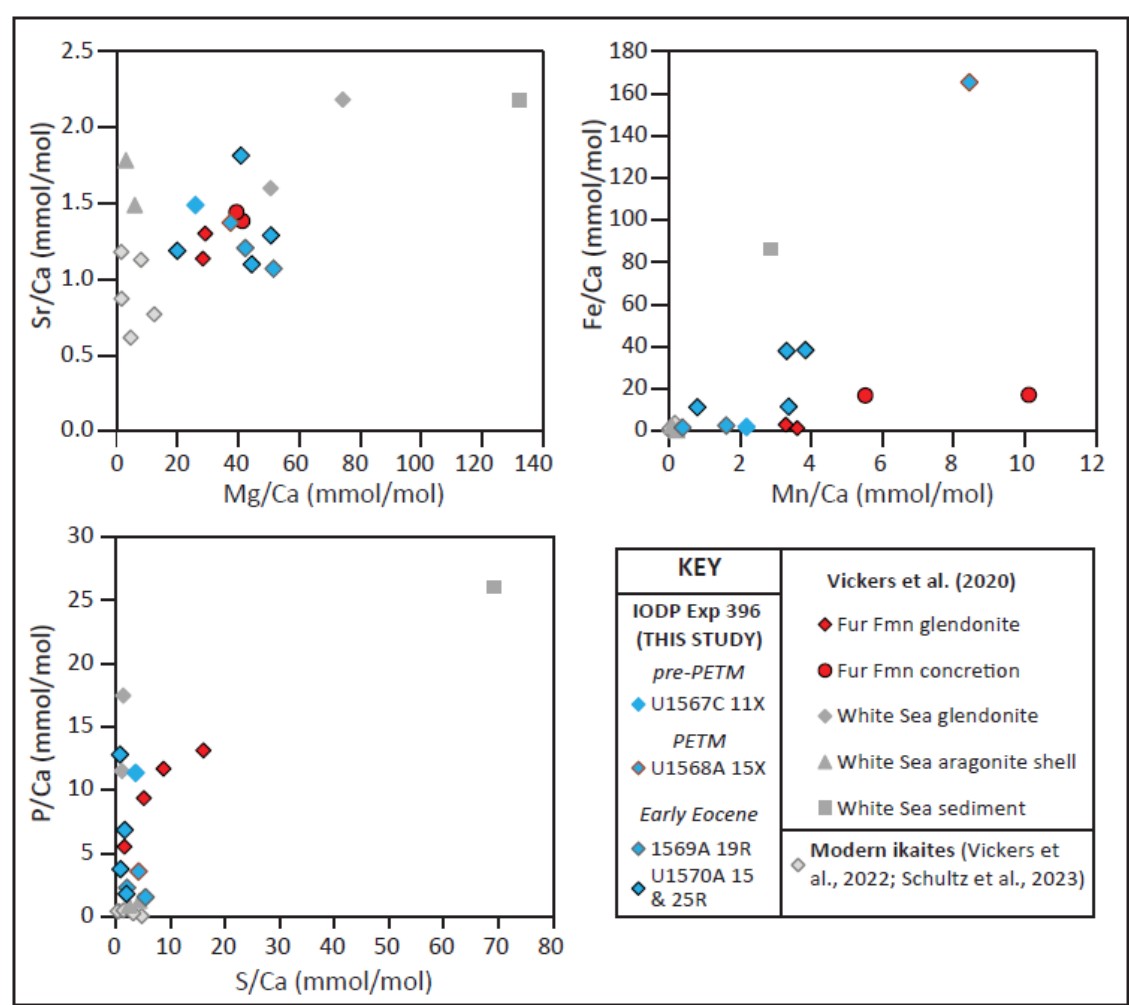

**Figure 7:** Element/Ca ratios of the Exp. 396 glendonites and associated calcites compared to published ICP-OES data for other glendonite-bearing sites.

1040

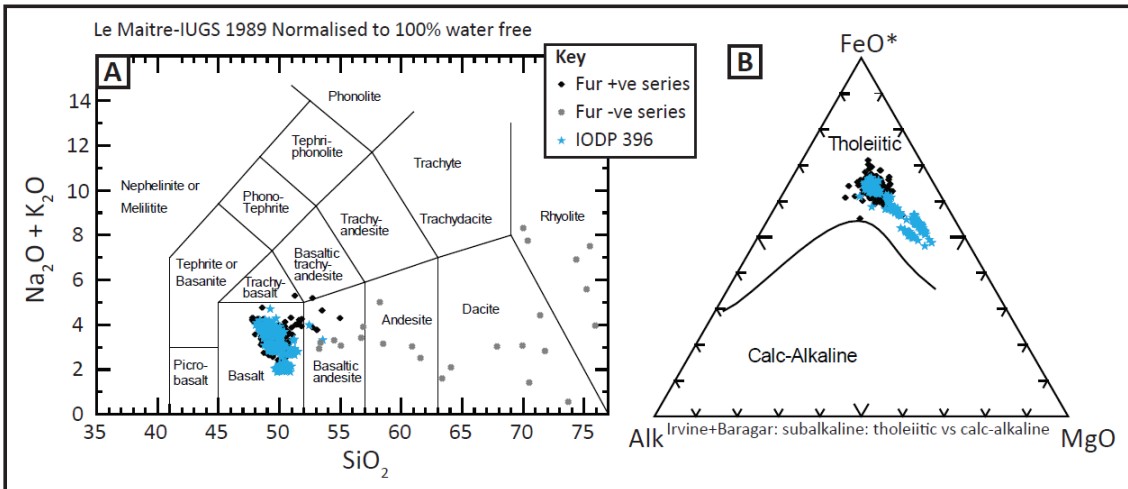

**Figure 8: (A)** A Total Alkali Silica (TAS) plot comparing the Exp. 396 ashes (this study) to published data for both positive (Stokke et al. 2020b) and negative (Larsen et al., 2003) ash series of the Fur Formation in northern Denmark. The Exp. 396 ashes and Fur positive series fall into the basaltic fields, whereas the Fur negative series show much more variation and have overall more felsic compositions. Note that while the Fur positive series data are microprobe analyses of matrix glass, the Fur Negative series data are whole rock data. However, the whole rock samples were leached of clay prior to analysis and no significant dilution is expected. **(B)** Ternary Alkali-Iron-Magnesium (AFM) diagram showing that the basaltic ashes from both the Exp. 396 sites and the Fur positive series are tholeiitic basalts. Note that many of the Exp. 396 ashes have higher MgO content than the Fur positive ashes.

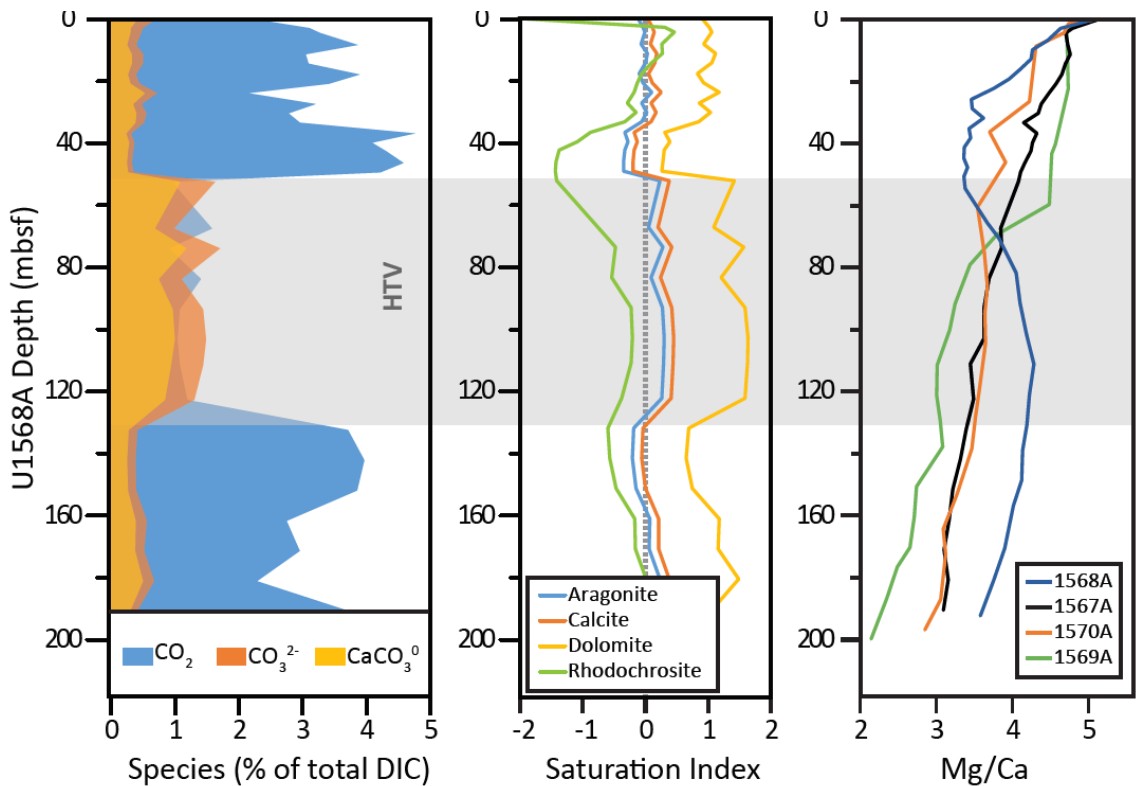

**Figure 9: (A)** PHREEQC simulation results for carbonate speciation in the U1568A core, which spans the hydrothermal vent infill (grey highlight labelled 'HTV'). Note that HCO3- (the major species) is not shown. **(B)** PHREEQC simulation saturation indices for several carbonate polymorphs. **(C)** IW Mg/Ca profiles for all glendonite-bearing cores (Planke et al., 2023b,c).

1042

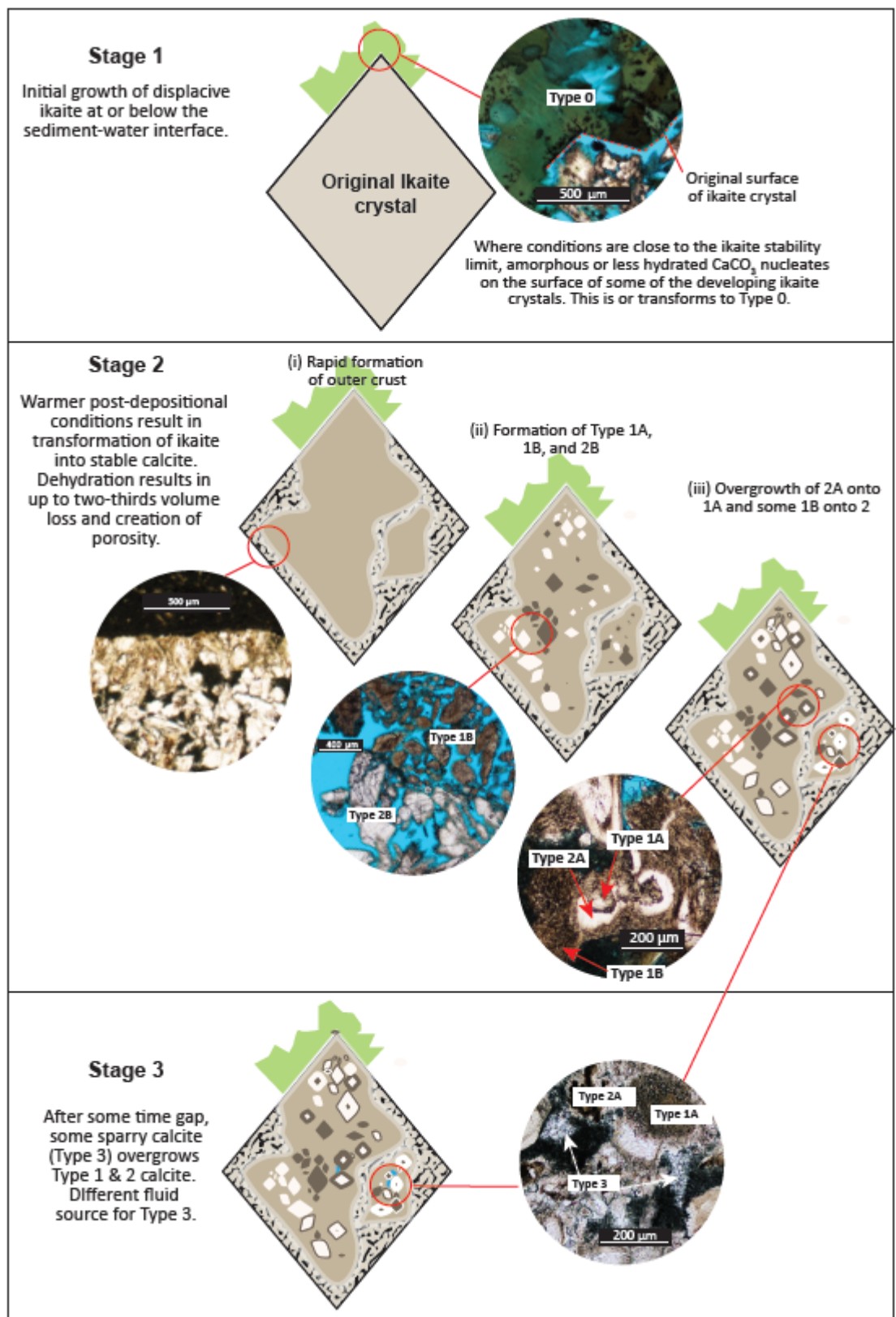

**Figure 10:** Schematic of ikaite transformation in the Exp. 396 cores, adapted from Counts et al.

(*accepted*) based on observed textural relationships and geochemistry of the calcite phases in the Exp.

396 glendonites.

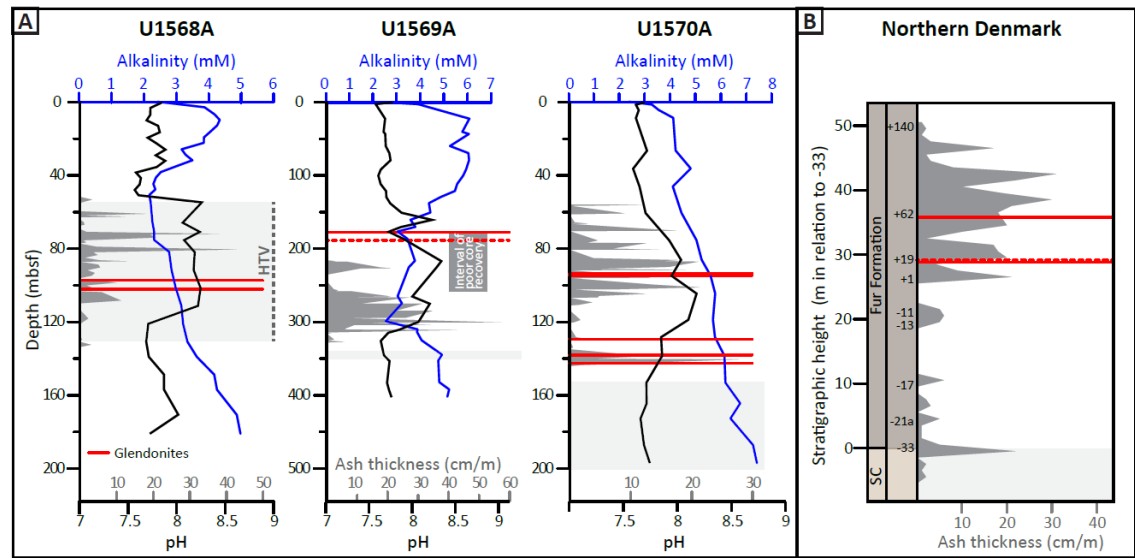

**Figure 11: (A)** Relative position of glendonites in the Paleocene-Eocene sediments of selected cores from the mid-Norwegian Margin, Exp. 396, compared to measured ash thicknesses. Pore water alkalinity and pH data (Planke et al., 2023) are also shown. Pale grey indicates the PETM-aged intervals in the stratigraphy. Note that for U1569A, core recovery was poor, particularly in the bottom, ash-bearing part (see Fig. 2). High ash contents lead to lower core recovery as they are course-grained and unlithified; therefore it is likely that there were much more numerous and thicker ash horizons in the interval between 18R and 37R (c. 180 – 340 mbsf). (B) Relative position of glendonites in the Paleocene-Eocene sediments of Northern Denmark, compared to ash thicknesses per metre (Jones et al., 2023). Glendonite horizons for the Fur Formation are from Vickers et al., (2020) (solid lines) and dashed line as identified by Henrik Friis, pers. comm. Pale grey indicates the end of the body of the PETM carbon isotope excursion (Jones et al., 2023). The recovery phase is between ashes -33 and -21a. SC = Stolleklint Clay.

| Site | Hole | Core | Core type | Section | Top depth (cm) | Bottom depth (cm) | mbsf top | PXRD mineralogy | Remarks |
|------|------|------|-----------|---------|----------------|-------------------|----------|-----------------|---------|
| 1567 | C | 10 | X | 3 | 40 | 45 | | | Displaced (in drill mud) |
| 1567 | C | 11 | X | 1 | 83 | 93 | 82.03 | Calcite, minor halite, Qz | Not cemented |
| 1568 | A | 15 | X | 1 | 29 | 31 | 97.3 | | Porous calcite in cement |
| 1568 | A | 15 | X | 4 | 49 | 51 | 102.0 | | Not cemented |
| 1568 | A | 15 | X | 4 | 55 | 58 | 102.1 | Calcite, minor Qz, rhodochrosite | Not cemented |
| 1569 | A | 19 | R | 2 | 54 | 62 | 177.3 | Calcite, Mg-calcite, minor Qz, gypsum, halite | Partially cemented |
| 1570 | A | 15 | R | 1 | 22 | 25 | 93.6 | | Not cemented.  Small fragment |
| 1570 | A | 15 | R | 1 | 108 | 112 | 94.5 | Calcite, minor Qz + halite | Not cemented, half glendonite |
| 1570 | A | 22 | R | 2 | 89 | 91 | 129.3 | | Cemented glendonite fragment (tip of crystal) |
| 1570 | A | 24 | R | 1 | 96 | 98 | 138.1 | | Uncemented glendonite fragments |
| 1570 | A | 25 | R | 1 | 50 | 55 | 142.5 | Calcite, minor Mg-calcite, Qz | Cemented (in lmst) |

**Table 1:** Glendonites of the Exp. 396 cores, PXRD data from bulk glendonite analysis, element/Ca ratios.

| Carbonate phase | Description |
|---|---|
| Type 0 | Green-brown, microcrystalline, granular carbonate phase, not observed in all glendonites (Fig. 4). Heterogeneous colour distribution from browner to greener areas, not visibly zoned under plane polarised light (Fig. 4B and F). Commonly shows black dendritic surface patterns. SEM imaging revealed Type 0 to be composed of micro-grains of carbonate, rather than being a single crystal (Fig. 5). |
| Type 1A | Forms dark and light brown zoned, rounded anhedral patches known as 'blebs' (Figs. 4E,F,H and 5A). The BSE SEM shows it has higher porosity than the calcite overgrowths (Fig. 5B), and EDS mapping shows some chemical zoning defined by its Mg concentration (Fig. 5F). Equivalent to Type 1 (I) of Huggett et al. (2005), Vickers et al. (2018), Schultz et al. (2023), and Counts et al. (2023). |
| Type 1B | Uneven coloured anhedral brown to dark brown calcite. May radially overgrow, or intergrow with, pale, non-porous Type 2A. Type 1B is indistinguishable from Type 1A except that it grows over Type 2A rather than the other way around, and generally has slightly higher [Mg] than 1A. Type 1B is not visibly zoned, and makes up larger patches/areas (Fig. 4 C – F; Fig. 5). May show cracking along cleavage planes (Fig. 4C). |
| Type 2A | White to pale brown, concentrically zoned calcite (under plane polarised light) that directly overgrows Type 1A, showing Mg-zoning (Fig. 4), and generally higher Mg than Type 1A (Fig. 5). Low/no porosity compared to Type 1A&B (Fig. 5). Equivalent to Type 2 (II) of Huggett et al. (2005), Vickers et al. (2018) and Schultz et al. (2023) and Type 2A of Counts et al. (2023). |
| Type 2B | Very similar to Type 2A; clear/white crystalline calcite with no porosity zoning. Unlike 2A, it also lacks chemical zoning, and may show cracks along cleavage (Fig. 5). Type 2B calcite is generally characterised by a higher Mg than Type 1A&B calcite phases (Fig 5). |
| Type 3 | Isopachous sparry or fibrous epitaxial calcite overgrowths to Types 1 and 2; higher [Mg] than types 1 and 2 (Figs. 4H and 5). |

**Table 2:** Descriptions of the different carbonate phases observed within the glendonites through thin section microscopic and geochemical analysis (light microscopy, SEM, EDS and LA-ICP-MS).