# Peer review of "Paleocene-Eocene age glendonites from the Mid-Norwegian Margin"

_EGUsphere, 2023_

## Author Comment (AC2)

**Response to RC1, Mikhail Rogov**

We are pleased that you find the MS by Vickers et al. an important contribution devoted to the enigmatic glendonite occurrences across the PETM climatic optimum, and that in general you find the manuscript well-written, with well-supported interpretations of the findings. In response to the specific line-by-line comments:

Line 83: "numerous glendonites in volcanic sediments" – rather, in "ash-bearing deposits"

- Indeed, ash-bearing deposits is a more accurate description, and we will make this replacement in the text.

Line 96: "glendonites in the stratigraphy of the Exp. 396 cores" – in my opinion, usage of the word 'stratigraphy' in such a meaning if doubtful; I propose to replace it by "glendonites recovered from Exp. 396 cores and their stratigraphic distribution"

- We agree, this way of phrasing is clearer and more accurate, and we will make this change.

Line 108: "The Modgunn locality is a transect of boreholes" – can the transect be considered as a 'locality'?

- We can rephrase this to "*Boreholes from* the Modgunn *transect* (Sites U1567-U1568) span the crater of a Paleogene hydrothermal vent complex". Likewise, when introducing the two transects, we can word this "Paleocene-Eocene sedimentary successions *were cored* along the Modgunn and Mimir *transects* (Fig. 1B and C)." to avoid any confusion.

Line 116: "and the biostratigraphic marker taxa Apectodinium augustum and Hemiaulus proteus" – rather, "FAD of the biostratigraphic marker taxa…". Can you show key biostratigraphic events (FAD and LAD of dinocyst species) on the figures? Intwill be useful for readers

- We will add the first and last occurrence data for Apectodinium augustum and Hemiaulus proteus (from Berndt et al., 2023) to Fig. 2, along with both published and new carbon isotope data for chemostratigraphic interpretation. We will edit this line to read "The strata contain a negative $\delta^{13}$C excursion and the *first and last occurrence of the* biostratigraphic marker taxa *Apectodinium augustum* and *Hemiaulus proteus* (*Fig. 2*)."

Line 228: "hydrothermal vent infilling sediments"- or "deposits" (here and above in the text)? the term 'sediments' more frequently used for modern unconsolidated ones

- We will replace "sediments" with "deposits" in both instances.

Line 242: at least a short review of biostratigraphic data, which are crucial for further discussion, is necessary prior the description of glendonites.

- We will add the published biostratigraphic from Berndt et al. (2023), along with carbon isotopic data (new and from Berndt et al., 2023) to Figure 2. We describe the biostratigraphic data briefly in the geological setting, but cannot describe it in the results as it is already published and discussed in Berndt et al., 2023, and not new data gathered for this study.

Lines 247-249: "Most show the characteristic shape of stellate or bladed 'crystals', although the individual blades are no longer a single crystal but rather a heterogeneous mix of smaller crystals" - as follow from photographs provided in the MS, nearly all recorded glendonite specimens can be

ascribed to a single rosette morphotype (following terminology proposed by Frank et al., 2008), except for specimens from figs. S4 and S8-S9, which morphology is unclear.

- We will add photos of all identified glendonites to Fig. 3 so that the readers do not have to refer to the supplementary material. We will change the sentence as recommended to *"Nearly all the recorded glendonite specimens can be ascribed to a single rosette morphotype (Fig. 3) (following terminology proposed by Frank et al., 2008), except for specimens from Fig. 3D, F, I and J, in which the morphology is unclear due to the fragmented nature or disturbance of the structure during drilling."*, and will add Frank et al. (2008) to the reference list.

Lines 267, 300, 320: "Counts et al., in review" it is not necessary to cite such a paper, which still not accepted yet; in all the cases it cited along with other refs

- We will refer instead to the Geological Society of America Annual Meeting Abstracts with Programs, 55, 6, which presents this model.

Lines 301-302: "green Type 0 calcite identified in this study has not been observed in other glendonite thin section" - can this newly recorded generation be related with an influence of the nearby ash horizons?

- It may be, although note that this phase has not been reported in the Fur Formation glendonites of northern Denmark (e.g., Vickers et al., 2020 Fig. 3), or indeed observed by the lead author in prepared thin sections of these glendonites, which are also associated with thick ash horizons. More investigation needs to be undertaken on this before a conclusive driver of the formation of this calcite phase can be identified.

Line 311: "localisedincrease" – please split this word

- Yes, we will put in the missing space.

Unfortunately, an information about the precise coordinates of studied boreholes is missing in the MS; I propose to add a table with these data to the Supplementary.

- We will add latitude and longitude information to section 3.1. where we describe from which holes the glendonites were recovered: "The glendonites discovered by IODP Expedition 396 are found within two holes of the Modgunn *transect*, U1567C *(3°3.219'E 65°21.785'N)* and U1568A *(3°3.109'E 65°21.594'N)*; *and within two holes of the* Mimir *transect*, U1569A *(2°1.608'E 65°49.878'N)* and U1570A *(1°59.623'E 65°49.890'N)* (Fig. 2 and Table 1)". Full information for all holes for Exp. 396 (coordinates, water depth, penetration, drilled interval etc) can be viewed in Table T1 of Planke et al. (2023a).

---

## Author Comment (AC3)

**Response to RC3, William Rush**

We thank William Rush for his considered comments on the manuscript. In addition to some minor comments, RC3 expressed their major concern that that more discussion of anaerobic oxidation of methane was required, as there is extensive literature on the formation of thermogenic methane in the North Atlantic during the PETM, the anaerobic oxidation of which could have influenced the formation of ikaite. This is a valid point, and we will accordingly add a discussion of the possible source and role of methane in ikaite precipitation to the manuscript (see response to comment on Lines 340-341 and Line 438. As the review lists the major and minor points in order they appear in the manuscript, we will answer the line-by-line comments in order also, in order to orient the reader.

Lines 48-49: specify °C when saying "temperatures approaching zero".

- We can add a bracket with the temperature as follows "*cold water* temperatures *(< 10 °C)* are much more favourable for the precipitation, growth, and longevity of this mineral."

Lines 62-63: explain why the glendonites are a topic of controversy, there should be more discussion on these previous studies that are referenced.

- We will clarify this by adding "have been a topic of controversy in the paleoclimate community *due to the apparent mismatch with concurrent paleotemperature proxies* (Huggett et al., 2005; Spielhagen and Tripati, 2009)"

Lines 78-80: "Whether these cooling events reflect a localised as well as short-lived cooling is not understood, nor the mechanism by which such cooling could occur, although these have been speculated upon" discuss how these have been speculated upon and what some of the proposed mechanisms have been.

- As the one of the main points of this paper is to discuss the evidence for and proposed mechanisms of cooling in this region, we would rather keep this as an introduction of the problem which is later discussed in detail (see section 4.3 'Timing and conceptual model for ikaite growth'). We propose to change this sentence to "*While the potential drivers of intermittent cooling episodes have been touched upon by previous studies (Schoon et al., 2015; Stokke et al., 2020a; Vickers et al., 2020), the causes, spatial extent, and recurrence times of such events remains unresolved.*"

Line 244: Somewhat minor, but I believe there could be a better word choice than "mush", perhaps mixture or amalgam?

- We will change the word "mush" to "*amalgam*".

Lines 296, 310: Again, I believe there is a better word choice than "blebs". I will leave that choice to the author, as I'm not exactly certain what is meant by blebs.

- We have added the following definition of the word to improve clarity: "Notably, areas of calcite blebs (bubble-like mineral inclusions)…"

Line 311: there is a space missing between "localised" and "increase".

- We will add this missing space in.

Line 340-341: This is my major concern with the paper. The authors reference anaerobic oxidation of methane as playing a key role in ikaite precipitation, yet do not discuss the extensive literature on the thermogenic methane production in the North Atlantic during the PETM associated with North

Atlantic Igneous Province vulcanism. There are many papers on this topic, including Jones et al., 2019 and Frieling et al., 2016 among others. At the very least, there should be discussion as to the possibility that anaerobic oxidation of methane could have played a role in the formation of ikaite as an alternative to the volcanic ash hypothesis, why this methane could not have played a role, or the potential that these mechanisms could have worked in conjunction.

- This is a good point, and we will accordingly expand the discussion to include the methane-ikaite link hypothesis. Here, we will add a section as follows: "The breakdown of sedimentary $C_{org}$ via sulphate reduction, and/or the anaerobic oxidation of methane (AOM) are thought to play a key role in ikaite precipitation, largely because the low $\delta^{13}C$ values measured in ikaites and glendonites suggest an organic or methanogenic source of carbon (e.g. *Rogov et al., 2023 and references therein*), and also because these organic matter decomposition processes generate DIC (*Hiruta and Matsumoto, 2022;* Whiticar et al. 2022). *Methane has been linked to ikaite/glendonite formation due to their frequent proximity to methane seeps (Greinert and Derkachev, 2004; Teichert and Luppold, 2013; Hiruta and Matsumoto, 2022) and gas inclusions containing methane and other hydrocarbons in glendonite specimens from the Jurassic of Siberia (Morales et al., 2017). However, other studies which examined sedimentary biomarker evidence for AOM in Oligocene and Eocene-aged glendonite-bearing strata did not find evidence for significantly elevated rates of AOM and support an organic matter source for the examined glendonites (Qu et al., 2017; Vickers et al., 2020).*" For ikaite to be precipitated over the more stable $CaCO_3$ polymorphs, factors inhibiting calcite and promoting ikaite precipitation are also required. These may include high alkalinity, high concentrations of phosphate and/or $Mg^{2+}$, and low temperatures (Rickaby et al., 2006; Zhou et al., 2015; Purgstaller et al., 2017; Stockmann et al., 2018). *The explosive NAIP* emplacement may have played a key role in generating *just* such conditions *for $CaCO_3$ precipitation and the precipitated polymorph being ikaite. Hydrothermal venting of methane and other gases occurred at sites proximal to the NAIP (Svensen et al., 2004; Frieling et al., 2016; Jones et al., 2019; Berndt et al., 2023); and the explosive nature of the NAIP eruptive volcanism could have driven short-term climate cooling ('volcanic winters', e.g. Robock et al., 2000; Schmidt et al., 2016; Stokke et al., 2020a). Furthermore, the large amounts of volcanic ash deposited in the sediments likely underwent rapid diagenesis, generating the chemical conditions in the pore waters that could have inhibited calcite precipitation and therefore promoted ikaite precipitation (e.g. Gislason and Oelkers, 2011; Olsson et al., 2014; Murray et al., 2018). Indeed, ikaite and other carbonates were discovered as travertine in the Hvanná river in the vicinity of the Eyjafjallajökull volcano shortly after eruptive activity began in Spring of 2010 (Olsson et al., 2014).*". We will further discuss the evidence for methane in the 396 cores and Nordic Seas region a little later on in the manuscript (see response to comment Line 438).

Line 393: there is a missing space between "CaCO3" and "in".

- We will add this missing space in.

Line 438: Again, here the influence of methane should be discussed in addition to the rapid diagenesis of ash.

- We will change this to "For sediments such as those encountered in Modgunn and Mimir, the primary driver of DIC *was likely AOM (rather than bacterial sulphate reduction of organic matter), since TOC in the sediments is low (generally < 1.5 wt %; Planke et al., 2023b,c) and there was much methane venting in the area (Svensen et al., 2004; Berndt et al., 2023)*".

Lines 449-450: A recent paper in this journal (Rush et al., 2023) found transient cooling associated with ETM2 in the Mid-Atlantic of the United States. This would not have been an enclosed basin but provides evidence of another region that experienced cooling during an Eocene hyperthermal and may be worth a citation.

- Very exciting. Also Meckler et al. 2022 found variable temperatures at odds with global for the North Atlantic. We will edit this section to include these new findings: "Paleotemperature estimates *for the PETM and early Eocene from both biomarkers and stable and clumped isotope thermometry for the North Atlantic and Nordic Seas region show variable temperatures a*nd intervals of cooling which are apparently at odds with global records *(Schoon et al., 2015; Stokke et al., 2020; Vickers et al., 2020; Meckler et al., 2022; Rush et al., 2023)*."

---

## Author Comment (AC4)

**Response to EC1, Gerilyn (Lynn) Soreghan**

We thank Gerilyn (Lynn) Soreghan for these helpful comments, and can address all the points raised, as follows:

117— the strata "exhibit" rather than "contain"

- We will make this word substitution ("The strata *exhibit* a negative $\delta^{13}C$ excursion…").

246— Section 3.2— it might be helpful to begin here by noting the size (and maybe shape) range, where you note they are variable in "size and appearance," in addition to the references to the relevant figures.

- We will elaborate the description of the glendonites as follows: "The glendonites are variable in size and appearance, with some being cemented or partially cemented (Fig. 3C, E, H and I), and some present as an uncemented *amalgam* of smaller crystals (Fig. 3B, D, F, G), and some retaining their structure but as a porous mesh of calcite (Fig. 3A and J). *In size, they range from small fragments (2 – 5 mm across, Fig. 3A,D,F,I,J) to crystals up to or beyond the than the entire width of the core (Fig. 3B,C)*. In some cases, the crystal appears to have grown over and incorporated parts of the host sediment (Fig. 3E,H), yet in others appears to have either displaced the sediment it was growing in (Fig. 3G), or grew up into the water column with later sedimentation burying it (Fig. 3C). N*early all the recorded glendonite specimens can be ascribed to a single rosette morphotype (Fig. 3) (following terminology proposed by Frank et al., 2008), except for specimens from Fig. 3D, F, I and J, in which the morphology is unclear due to the fragmented nature or disturbance of the structure during drilling.*"

Fig. 3A illustrates a core that appears to have mud invasion and be a bit disrupted, and indeed the caption notes that these are glendonite fragments in drill mud. Doesn't that mean that these could be out of place, and thus not reliably placed stratigraphically? In other words, why show this example, given the uncertainty of its stratigraphic position? I see that you mention this in lines 257-260, so that is fine.

- We also describe this in Table 1 in the 'remarks' column.

315— space needed between words.

- we will add this missing space ("localised increase").

367— what is meant by "Pore waters… were taken on board at low resolution…"— really two questions— how were the pore waters sampled, and at what resolution? Did I miss this in the Methods? (I couldn't find this detailed in the methods).

- Shipboard sampling and analyses of the interstitial waters were made during the Expedition 396, and the methods and results are published in Planke et al., 2023 a, b and c. We will delete this line from the discussion as it pertains to methods, and add a paragraph to section 2.1 ('Geological setting') and sampling to explain this as follows:

"*At each Hole, interstitial water (IW) samples were taken at intervals of ~3 m of sediment in the upper 50 metres, ~1 sample every 9.5 m for the lower parts of the cored sediment. Standard IODP methods for IW extraction were used at all sites. Following whole-core recovery to the catwalk, full round samples were collected, sealed and transferred to the shipboard chemistry lab, where sediment exteriors were carefully removed to reduce potential contamination from drilling fluids. The samples were individually 'squeezed' - placed into a Carver press and subjected to 35,000 lb force.*

*Squeezed fluid was then filtered through a Whatman No. 1 filter (11 µm) and 0.5 mL was discarded. The remaining fluids were collected in acid-cleaned syringes after filtering through 0.45 µm polyethersulfone membranes, and split into aliquots. All analyses of the collected IWs were completed following the standard shipboard methods of the R/V 263 JOIDES Resolution (Planke et al., 2023d), and are published in Planke et al. (2023a) and (2023b)."*

495— are they found "throughout" or within discrete intervals? It seems more like the latter?

- The fact that so many glendonites were sampled in numerous intervals within the narrow boreholes from the expedition 396 suggests that there are likely many more glendonite horizons which were missed, and therefore feel that the word "throughout" is more apt.

---

## Author Comment (AC5)

**Response to CC1, Niels de Winter**

We thank Niels de Winter for his thoughtful community comment, and are happy to address every point raised. We believe that the changes we will make in response make the manuscript a much more thorough and considered discussion on the conditions that may have driven glendonite formation throughout the Paleocene-Eocene-aged sediments from the mid-Norwegian margin (Exp. 396).

*General comments*

The major issues CC1 raised were that 'the observations and data the authors put forward do not conclusively support the hypothesis that the Eocene hothouse was punctuated by geologically brief "cold snaps".' And proposed that 'the lack of precise timing of the formation of the glendonites leaves room for other explanations, such as the hypothesis that the fast-growing glendonites form out of isotopic equilibrium (and may record lower temperatures than their growing temperatures) or that they grow seasonally (e.g. during the cold season).'

We point out that whether or not the parent ikaite to the glendonites grew in or out of equilibrium, the fact remains that ikaite is more stable at low temperatures. If the ikaite s grew very rapidly, as we believe they did, this is more likely to require very low temperatures, as calcite or other $CaCO_3$ polymorphs are favoured over ikaite as temperatures increase (e.g., recent findings for the Ikka Fjord glendonite tufas find that since the fjord waters have warmed several degrees, $CaCO_3.H_2O$ is precipitating on top of the older ikaite, Tollefsen et al., 2022). We have included mention of possible disequilibrium effects relating to the previous study which presents glendonite clumped isotopes (see detailed response to comments below), and further experiments on ikaite precipitation will be undertaken in future work; however as this manuscript does not present any isotopic data (e.g., stable, clumped, or dual clumped isotopes), extensive discussion of the caveats surround clumped isotope temperatures are beyond the scope of the paper. We also respond briefly to the comment about seasonal bias in different proxies (again, see detailed responses to comments).

CC1 also suggested that the PHREEQC modelling is the weakest part of the manuscript, and that they were sceptical about the use of pore water chemistry in Eocene sediments to infer something about paleo-conditions. They suggested to dexcribe the outcomes of the PHREEQC model in the results, and requested that the authors elaborate a bit more on these results and how they add to the discussion in this manuscript. We will add a section in the results detailing the PHREEQC model outcomes and describing the pore water profile trends, and discuss the implications with respect to laboratory-based ikaite synthesis experiments at warm temperatures (> 10 C) (see detailed responses below).

*Detailed (line-by-line) responses to minor and major comments*

Line 70: "…the conditions under which this was achieved in the laboratory is unlike any natural setting." should probably read "…the conditions under which this was achieved in the laboratory are unlike any natural setting."

-   Accordingly, we have change "is" to "are"

Line 76-78: On checking the preprint by Jones et al. (2023), most specifically figures 3 and 6, I did not find any biomarker-based temperature reconstructions that yielded results below 10°C. It seems the only datapoints in this compilation that yield colder temperatures are those originating from Vickers et al. (2020), which represent clumped isotope datapoints on glendonites if I'm not mistaken. Unless I have overlooked any temperature reconstructions in Jones et al. (2023) the authors are referring to

here, I think this statement should be rephrased. The authors should acknowledge that the cold temperature reconstructions are only found from analyses on the glendonites themselves, and not from other (independent) proxies and archives. This observation has implications for the "cold snap" hypothesis, as it remains possible that the temperatures reconstructed from the glendonites themselves are underestimations (see comments on lines 441-462).

- We should have been clearer here: We were talking about the *magnitude* of cooling events being of 5 -7 C – the absolute SSTs were not much below 15C during these events. We have changed this to read: "Reconstructed sea surface temperatures from biomarkers for northern Denmark also suggest that short-term cooling events of magnitude c. 5 - 7 °C (*down to SSTs of 15 °C or lower*) may have punctuated the late Paleogene to early Eocene (*Stokke et al., 2020a; Vickers et al., 2020*; summarised by Jones et al., 2023)." (note that Jones et al., 2023 has now been published in Clim. Past.)

Line 92-94: The train of thought in this sentence is a bit hard to follow for me. Perhaps the authors could briefly explain how these colder temperature reconstructions from glendonites imply changes in circulation or stratification in this basin. I'm sure this has been discussed in one or more of the papers by the authors cited earlier in the Introduction, but for the sake of clarity I would suggest the authors indulge the reader who has not read these contributions in detail by explaining the line of reasoning here and refer to these previous studies for a more detailed discussion.

- We will change this to: "If the parent ikaite grew during the PETM or the hothouse earliest Eocene climate (as was the case for the well-studied Fur Formation glendonites; Vickers et al., 2020), this raises questions about regional seaway connectivity and thermal stratification in both the Nordic Seas and the open oceans, as an extreme thermocline within these shallow seas is one way to reconcile cold bottom water and warm surface water proxies."

Section 2.2-2.6: I commend the authors on their detailed explanation of the geochemical measurement procedures.

- Thank you!

Section 3.7: It would probably benefit the manuscript if the results of PHREEQC modelling are described in more detail here. As it stands, this section now contains one sentence referring to a figure and supplement, which leaves the reader searching for the information.

- This section (now 3.9) has been expanded to give some background information on why the interstitial waters are somewhat anomalous, including references to interstitial water data from Planke et al. (2023): "*The shipboard analyses of interstitial water (IW) samples showed non-typical behaviour within Units IV and V (the vent infill) at Site 1568A, with lower alkalinity (2–3 mM) and higher pH (~8.2) compared to both underlying and overlying strata (Planke et al., 2023b). Many IW profiles in ocean sediments above igneous basement show a marked reduction in dissolved Mg/Ca ratios with depth (e.g. the infill interval (51.45–119.09 metres below surface, mbsf) shows an inversion in dissolved Mg/Ca ratios to increasing values with depth (Fig. 9).* Results from the PHREEQC modelling for the interstitial waters (IW) are shown in Fig. 9 and in the supplementary data. *Dissolved inorganic carbon (DIC) speciation is shifted towards [CO32-] across the HTV infill and as a result the saturation index of the CaCO3 minerals increases in these horizons, in the case of calcite to values greater than zero.*"

Line 263-269: While I understand the decision to keep the terminology of the different carbonate phases observed within the glendonites consistent with previous literature, I think it would be helpful to include a brief description of the crystal habit (e.g. "botryoidal", "sparry", etc.) and texture of each phase (or "type") here in the Results section.

Descriptions of the crystal habit, as much as possible, are given in table 2, to which we will improve the clarity of some of the descriptions as follows:

| Carbonate phase | Description |
|---|---|
| Type 0 | Green-brown, *microcrystalline, granular* carbonate phase, *not observed in all glendonites* (Fig. 4). Heterogeneous colour distribution from browner to greener areas, not visibly zoned under plane polarised light (Fig. 4B and F). Commonly shows black dendritic surface patterns. SEM imaging revealed Type 0 to be composed of micro-grains of carbonate, rather than being a single crystal (Fig. 5). |
| Type 1A | Forms dark *and light brown zoned, rounded anhedral patches known as 'blebs' (Figs. 4E,F,H and 5A)*. The BSE SEM shows it has higher porosity than the calcite overgrowths (Fig. 5B), and EDS mapping shows some chemical zoning defined by its Mg concentration (Fig. 5F). Equivalent to Type 1 (I) of Huggett et al. (2005), Vickers et al. (2018), Schultz et al. (2023), and Counts et al. (2023). |
| Type 1B | Uneven coloured *anhedral* brown to dark brown *calcite. May radially* overgrow, or intergrow with, pale, non-porous Type 2A. Type 1B is indistinguishable from Type 1A except that it grows over Type 2A rather than the other way around, and generally has slightly higher [Mg] than 1A. Type 1B is not visibly zoned, and makes up larger patches/areas (Fig. 4 C – F; Fig. 5). *May show cracking along cleavage planes* (Fig. 4C). |
| Type 2A | White *to pale brown, concentrically zoned* calcite (under plane polarised light) that directly overgrows Type 1A, showing Mg-zoning (Fig. 4), and generally higher Mg than Type 1A (Fig. 5). Low/no porosity compared to Type 1A&B (Fig. 5). Equivalent to Type 2 (II) of Huggett et al. (2005), Vickers et al. (2018) and Schultz et al. (2023) and Type 2A of Counts et al. (2023). |
| Type 2B | Very similar to Type 2A; *clear/white crystalline calcite* with no porosity zoning. Unlike 2A, it also lacks chemical zoning, and may show cracks along cleavage (Fig. 5). Type 2B calcite is generally characterised by a higher Mg than Type 1A&B calcite phases (Fig 5). |
| Type 3 | Isopachous sparry or fibrous epitaxial calcite overgrowths to Types 1 and 2; higher [Mg] than types 1 and 2 (Figs. 4H and 5). |

Line 296: Is "blebs" a scientific term in this context? Perhaps the authors can define it here for clarity.

- We will add the clarification "*bubble-like mineral inclusions*" here, and In table 2 we will add a line defining it as follows: "Forms dark *and light brown zoned, rounded anhedral patches known as 'blebs' (Figs. 4E,F,H and 5A)*".

Line 311: A space is needed between "localized" and "increase".

- We have accordingly put in this missing space.

Line 313: "due to the rapidity of the reaction" This reasoning requires a bit more explanation, I think. Do the authors mean that the faster mineralization rate causes less discrimination against Mg in the

crystal structure compared to phase 1A? If so, it would be helpful if the authors supported this claim about the influence of reaction rate with a reference.

- Good point. We have accordingly added a reference to Schultz et al. (2023), where the difference in fabric related to rate of transformation can been seen in the images shown in their Figure 3, and have clarified the text to read: "The formation of 1A blebs began during the recrystallisation reaction, preferentially excluding Mg from the crystal structure, leading to a highly localised increase of Mg2+ in the pore waters. *This may have taken place over timescales of years (Schultz et al., 2023b)*. Where breakdown was rapid, larger areas of Type 1B formed (*also observed in ikaite transformed on timescales of hours; Schultz et al., 2023b). The faster mineralisation rate of 1A compared to 1B is believed to have caused less discrimination against Mg in the crystal structure, as evidenced by the higher Mg content of 1B compared to 1A (Fig.6)*."

Line 339-344: Here I think the authors should acknowledge that this link between methane seepage and ikaite formation (and the appearance of glendonites in the sedimentary record) has also been recorded in geological history (e.g. Morales et al., 2017)

- We will change the reference to the review paper of Rogov et al., 2023, which presents a detailed discussion of the evidence for methane / org C as carbon sources for ikaite/glendonite, including Morales et al., 2017. We will expand the discussion of the methane-ikaite link as follows: "The breakdown of sedimentary $C_{org}$ via sulphate reduction, and/or the anaerobic oxidation of methane (AOM) are thought to play a key role in ikaite precipitation, largely because the low $\delta^{13}$C values measured in ikaites and glendonites suggest an organic or methanogenic source of carbon (e.g. *Rogov et al., 2023 and references therein*), and also because these organic matter decomposition processes generate DIC (*Hiruta and Matsumoto, 2022;* Whiticar et al. 2022). *Methane has been linked to ikaite/glendonite formation due to their frequent proximity to methane seeps (Greinert and Derkachev, 2004; Teichert and Luppold, 2013; Hiruta and Matsumoto, 2022) and gas inclusions containing methane and other hydrocarbons in glendonite specimens from the Jurassic of Siberia (Morales et al., 2017). However, other studies which examined sedimentary biomarker evidence for AOM in Oligocene and Eocene-aged glendonite-bearing strata did not find evidence for significantly elevated rates of AOM and support an organic matter source for the examined glendonites (Qu et al., 2017; Vickers et al., 2020)*."

Line 394-397: I find it hard to see the connection between PHREEQC model results and the discussion in this section, but I think that is mostly because the results of the modelling have not been described in the manuscript. It would be easier for the reader to follow the reasoning here if the outcomes of PHREEQC simulations are first described (in Results) and then discussed earlier in the discussion before they are used to support the discussion here.

- We will now expand the description of the PHREEQC output in the results section as follows: "*The shipboard analyses of interstitial water (IW) samples showed non-typical behaviour within Units IV and V (the vent infill) at Site 1568A, with lower alkalinity (2–3 mM) and higher pH (~8.2) compared to both underlying and overlying strata (Planke et al., 2023b). Many IW profiles in ocean sediments above igneous basement show a marked reduction in dissolved Mg/Ca ratios with depth (e.g. the infill interval (51.45–119.09 metres below surface, mbsf) shows an inversion in dissolved Mg/Ca ratios to increasing values with depth (Fig. 9).* Results from the PHREEQC modelling for the interstitial waters *(IW)* are shown in Fig. 9 and in the supplementary data. *Dissolved inorganic carbon (DIC) speciation is shifted towards [$CO_3^{2-}$]*

*across the HTV infill and as a result the saturation index of the CaCO₃ minerals increases in these horizons, in the case of calcite to values greater than zero.*".

Lines 399-440: I think the hypothesis for the formation of ikaite in connection to the deposition of the ashes in this section is very plausible.

- Excellent

Lines 441-462: I'm not sure if I am fully convinced that the data presented in this study and the previous studies cited here are conclusive evidence pointing towards "cold snaps" during the Eocene hothouse. In my opinion, there remain two other possibilities that could reconcile the cool temperatures measured in the glendonites using clumped isotope thermometry with the warmer temperatures inferred from biomarkers.

Firstly, it is possible that the transformation of ikaite to glendonite does not take place in (clumped) isotopic equilibrium. Studies using the comparatively new "dual clumped" method highlight that some carbonates (e.g. those in brachiopod shells or corals) are affected by rate-limiting processes such as the hydration of CO2 in the water and the diffusion of DIC to the mineralization site (Davies et al., 2023). If glendonites indeed grow as fast as hypothesized in the previous section (lines 427-433), these processes may also cause kinetic effects in their isotopic composition. In brachiopods, such effects are demonstrated to cause an offset between the D47-based temperature and the actual environmental temperature under which the carbonate mineralizes of up to 10 degrees, enough to potentially explain a large part of the temperature offset cited here (Bajnai et al., 2020; Davies et al., 2023). While I would not suggest the authors add dual clumped measurements of their glendonites to this study (which might be analytically challenging considering their heterogeneity in terms of carbonate phases), I think this caveat should be recognized as part of the discussion here.

- This is true; the kinetics of ikaite growth/transformation have yet to be fully explored. This is something for future study; in this study we do not present any clumped isotope data; we document and describe the glendonites found in the succession. However, we stress that the presence of glendonites alone implies low temperatures due to the temperature dependency of ikaite metastability, and that ikaite has never been found in nature growing at temperatures of > 10 °C, and the key arguments presented here are based as much on the presence of ikaite as the clumped isotope temperatures. Yet of course there is a possibility that they formed at higher temperatures, as implied by laboratory studies that managed to fleetingly precipitate ikaite at temperatures of > 10 C (Purgstaller et al., 2017; Tollefsen et al., 2020). We will clarify the statement in the introduction by specifying the difference between laboratory experiments and natural ikaite precipitation sites: "Whilst the successful synthesis of ikaite at warm (≤ 35 °C) temperatures in laboratory conditions raises the possibility of ikaite/glendonite formation at much warmer temperatures than modern-day natural ikaites (*Purgstaller et al., 2017*; Tollefsen et al., 2020), the conditions under which this was achieved in the laboratory are unlike *marine* natural settings (*e.g., compared to modern marine ikaite-bearing sites those precipitation experiments were characterised by DIC concentrations at least an order of magnitude higher than typical pore water profiles and* $\Omega_{calcite}$ *>100, far in excess of that typically found in natural settings; Zhou et al., 2015*)". Then in the discussion look at the remnant alkalinity and Mg/Ca of the pore waters, and MgO content of the ashes, to argue that the conditions reached in laboratory synthesis of ikaite at > 10 C are unlikely to have been met in these sediments, "*While this overall indicates that the conditions necessary for ikaite formation prevailed in the HTV infill, we note that the carbonate chemistry of the remnant pore water profiles is characterised by conditions that*

*are far less saturated with respect to CaCO₃ minerals than those necessary to form ikaite in the laboratory, especially at higher temperatures (Purgstaller et al., 2017; Tollefsen et al., 2020).*". Thus, although we cannot in this study test the kinetics of ikaite growth/transformation (as we are not presenting clumped or dual clumped isotope data), we argue that it is more likely that the parent ikaite grew at low temperatures, even if the clumped isotope data presented for the Fur Formation glendonites should prove, in a hypothetical future study, to reflect kinetic disequilibrium. We will also add a brief description of the caveats of the various proxies to the discussion as follows: "*Alternatively, however, these discrepancies may reflect biases in the specific proxies used, e.g., seasonal (Jia et al., 2017; Udoh et al., 2022; de Winter et al., 2021), or even issues with the proxies themselves; e.g., brGDGTs measured for biomarker-based temperature reconstructions may be affected by changing conditions or source (Zhang et al., 2016; Inglis et al., 2019), or, in the case of carbonate clumped isotopes, solid-state reordering pushing the apparent temperature up (Henkes et al., 2014), or kinetic disequilibrium during the carbonate precipitation (Daëron et al., 2019), meaning that these signals may be challenging to interpret as reflecting true aquatic temperatures at the time of formation in some settings.*"

Secondly, I think the authors should consider the possibility that the difference between the temperatures recorded in the glendonites and those in the organic proxies can be explained by a difference in the season in which these materials are formed. If the glendonites formed in the winter and the biomarkers represent a summer signal, it is not inconceivable that the former yields temperatures of 1-9 degrees and the latter 20-30 degrees, especially in the higher latitudes. In fact, similar seasonal temperature contrasts are common in modern marine systems like the North Sea (e.g. Van Aken, 2008) and have been observed in seasonal-scale temperature reconstructions from higher mid-latitudes in the same region during past greenhouse periods (e.g. de Winter et al., 2021). In addition, while the season of growth of recrystallization of the glendonites would be hard to constrain, at least there is some evidence that biomarker-based SST reconstructions may be seasonally biased (Jia et al., 2017; Udoh et al., 2022). It seems plausible to me that the ikaites form in the winter season when temperatures drop enough for the mineralization to start, even if the chemical conditions that seed ikaite formation (made possible by the volcanic ashes) are in place earlier in the year. Therefore, I think this hypothesis cannot be disregarded in this discussion.

- We agree, in fact our model is based on the assumption that there is a seasonality effect; a particularly cold, windy winter (e.g. driven by volcanic cooling) would cool surface water, which eventually lead to a cascade event whereby the denser cold water at the surface would sink below the warmer bottom water. After this, the basin would once more become stratified, with the cold, dense water remaining at the bottom and enabling ikaite to grow. Such a seasonal winter bias still needs an anomalously cold winter in this sub-Polar region. We will add the following to the manuscript "Without sufficient exchange with the global ocean, bottom waters colder than the global deep ocean, that may have formed during transient cool conditions such as *anomalously cold (e.g., volcanic-driven) winters*".

Finally, the authors make a link between crystal size and growth rates (lines 427-433). This allows room for the argument that relatively large (cm-scale) crystals can grow in relatively short cold periods. However, even considering the fast mineralization rates observed for ikaite, I wonder how the very large (up to meter-scale) in the Fur formation can form in a setting that is otherwise indicative of typical warm Eocene hothouse conditions (e.g. Schultz et al., 2020). Even assuming growth rates in the order of centimeters per year, growing such crystals would still require decades to centuries of cold periods in this area. While this observation seems to favor the "cold snap"

hypothesis rather than the seasonal growth I suggested as an alternative above, it still seems hard to explain such prolonged cool periods of which no evidence is present in the local geological record beyond the glendonites. This makes me think that perhaps the thickness or composition of the ash layers across the basin might be a factor influencing the differences in glendonite size between the locations (with the larger glendonites begin found further south). Perhaps the authors could comment on this in the manuscript.

- The glendonites are only associated with the thick, tholeiitic ashes in both the Fur Formation and the mid-Norwegian margin, as discussed in section 4.2. It is curious that the largest glendonites ever found are from the Fur Formation of Northern Denmark, some 700 kilometers away from the NAIP nearest volcano (Stokke et al., 2020 *Volcanica*; much further than the 396 drill sites in the Modgunn and Mimir transects), and where there is a lack of evidence for methane venting/seeping (e.g., Vickers et al., 2020; Jones et al., 2019). Further than this we cannot comment on, as the unusual large size of the Fur Formation "mega- "glendonites remains a mystery, beyond the scope of this study.